# Increased DNA methylation variability in type 1 diabetes across three immune effector cell types

Dirk S. Paul[1,2,*], Andrew E. Teschendorff[3,4,*], Mary A.N. Dang[5,*], Robert Lowe[5,*], Mohammed I. Hawa[5], Simone Ecker[1], Huriya Beyan[5], Stephanie Cunningham[5], Alexandra R. Fouts[6], Anita Ramelius[7], Frances Burden[8,9], Samantha Farrow[8,9], Sophia Rowlston[8,9], Karola Rehnstrom[8,9], Mattia Frontini[8,9,10], Kate Downes[8,9], Stephan Busche[11,12], Warren A. Cheung[11,12], Bing Ge[11,12], Marie-Michelle Simon[11,12], David Bujold[11,12], Tony Kwan[11,12], Guillaume Bourque[11,12], Avik Datta[13], Ernesto Lowy[13], Laura Clarke[13], Paul Flicek[13], Emanuele Libertini[1], Simon Heath[14,15], Marta Gut[14,15], Ivo G. Gut[14,15], Willem H. Ouwehand[8,9,10,16], Tomi Pastinen[11,12], Nicole Soranzo[8,16], Sabine E. Hofer[17], Beate Karges[18,19], Thomas Meissner[19,20], Bernhard O. Boehm[21,22,23], Corrado Cilio[7], Helena Elding Larsson[7], Åke Lernmark[7], Andrea K. Steck[6], Vardhman K. Rakyan[5,*], Stephan Beck[1,*] & R. David Leslie[5,*]

The incidence of type 1 diabetes (T1D) has substantially increased over the past decade, suggesting a role for non-genetic factors such as epigenetic mechanisms in disease development. Here we present an epigenome-wide association study across 406,365 CpGs in 52 monozygotic twin pairs discordant for T1D in three immune effector cell types. We observe a substantial enrichment of differentially variable CpG positions (DVPs) in T1D twins when compared with their healthy co-twins and when compared with healthy, unrelated individuals. These T1D-associated DVPs are found to be temporally stable and enriched at gene regulatory elements. Integration with cell type-specific gene regulatory circuits highlight pathways involved in immune cell metabolism and the cell cycle, including mTOR signalling. Evidence from cord blood of newborns who progress to overt T1D suggests that the DVPs likely emerge after birth. Our findings, based on 772 methylomes, implicate epigenetic changes that could contribute to disease pathogenesis in T1D.

[1] Medical Genomics, UCL Cancer Institute, University College London, London WC1E 6BT, UK. [2] Cardiovascular Epidemiology Unit, Department of Public Health and Primary Care, University of Cambridge, Strangeways Research Laboratory, Cambridge CB1 8RN, UK. [3] CAS Key Lab of Computational Biology, CAS-MPG Partner Institute for Computational Biology, Shanghai Institute for Biological Sciences, Chinese Academy of Sciences, Shanghai 200031, China. [4] Statistical Cancer Genomics, UCL Cancer Institute, University College London, London WC1E 6BT, UK. [5] The Blizard Institute, Barts and The London School of Medicine and Dentistry, Queen Mary University of London, London E1 2AT, UK. [6] Barbara Davis Center for Childhood Diabetes, University of Colorado School of Medicine, Aurora, Colorado 80045, USA. [7] Department of Clinical Sciences, Lund University, Skåne University Hospital, SE-20502 Malmö, Sweden. [8] Department of Haematology, University of Cambridge, Cambridge Biomedical Campus, Cambridge CB2 0PT, UK. [9] National Health Service Blood and Transplant, Cambridge Biomedical Campus, Cambridge CB2 0PT, UK. [10] British Heart Foundation Centre of Excellence, Cambridge Biomedical Campus, Cambridge CB2 0QQ, UK. [11] Department of Human Genetics, McGill University, Montreal, Québec, Canada H3A 0G1. [12] McGill University and Genome Quebec Innovation Centre, Montreal, Québec, Canada H3A 0G1. [13] European Molecular Biology Laboratory, European Bioinformatics Institute, Wellcome Genome Campus, Hinxton, Cambridge CB10 1SD, UK. [14] CNAG-CRG, Centre for Genomic Regulation, Barcelona Institute of Science and Technology (BIST), Baldiri Reixac 4, 08028 Barcelona, Spain. [15] Universitat Pompeu Fabra, Plaça de la Mercè 10, 08002 Barcelona, Spain. [16] Human Genetics, Wellcome Trust Sanger Institute, Wellcome Genome Campus, Hinxton, Cambridge CB10 1SA, UK. [17] Department of Pediatrics, Medical University of Innsbruck, 6020 Innsbruck, Austria. [18] Division of Endocrinology and Diabetes, RWTH Aachen University, 52074 Aachen, Germany. [19] German Center for Diabetes Research (DZD), 85764 Neuherberg, Germany. [20] Department of General Pediatrics, Neonatology and Pediatric Cardiology, University Children's Hospital, Heinrich Heine University of Düsseldorf, 40225 Düsseldorf, Germany. [21] Division of Endocrinology, Department of Internal Medicine I, Ulm University Medical Centre, 89081 Ulm, Germany. [22] Lee Kong Chian School of Medicine, Nanyang Technological University, Singapore 636921, Singapore. [23] Imperial College London, London SW7 2AZ, UK. * These authors contributed equally to this work. Correspondence and requests for materials should be addressed to D.S.P. (email: dsp35@medschl.cam.ac.uk) or to R.D.L. (email: r.d.g.leslie@qmul.ac.uk).

Type 1 diabetes (T1D) is a common, organ-specific autoimmune disease that results from the progressive loss of insulin-producing β-cells in the pancreas. Genetic predisposition and environmental factors contribute to the disease onset[1]. The incidence of T1D has dramatically increased in recent years (3–4% per annum), with the most rapid upsurge seen in children younger than five years of age[2]. The increasing rate of T1D, along with disease discordance in monozygotic (MZ) twins, suggest that non-genetic factors play a major role[3,4]. Such factors, including viral and bacterial infections, diet, and potentially epigenetic and stochastic events, may affect disease predisposition either *in utero* or in early childhood when predictive autoantibodies emerge[3]. However, conclusive evidence about causal environmental factors in T1D pathogenesis has not been obtained to date.

Epigenetic modifications, including DNA methylation, are cell type-specific and induce stable changes in gene expression that are heritable during cell division. DNA methylation occurs at cytosine residues mainly in the context of CpG dinucleotides, and is generally associated with transcriptional silencing[5]. It can contribute to disease development and progression through its influence on gene expression, and function as mediator in response to environmental stimuli[6]. In systematic epigenome-wide association studies (EWASs), DNA methylation levels are typically measured at hundreds of thousands of CpG sites across individuals in a case-control cross-sectional cohort. CpG sites are then associated with disease status, and differences in DNA methylation levels between cases and controls are recorded[7,8]. However, it has to be noted that the meaningful interpretation of EWAS findings is impeded by several confounding factors, in particular cellular heterogeneity in accessible sample material (for example, peripheral blood) and genetic heterogeneity between individuals[7,9].

Disease-associated CpG sites can be identified using different analytical approaches (Fig. 1). Most EWASs have sought to quantify differences in mean DNA methylation at CpG sites between cases and controls, that is, differentially methylated CpG positions (DMPs). In recent years, DMPs have been shown to associate with a multitude of complex traits and diseases, including blood pressure[10], triglyceride levels[11], pain sensitivity[12], schizophrenia[13], rheumatoid arthritis[14] and T1D (refs 15,16). However, the difference in mean DNA methylation at these CpGs is often small (<5%), raising challenges to their biological interpretation.

In parallel, the potential importance of increased DNA methylation variability has been noted in cancerous tissue[17–22]. Differentially variable CpG positions (DVPs) are heterogeneous outlier events that occur mainly, if not exclusively, in disease cases (Fig. 1). DVPs usually involve larger shifts in DNA methylation (>10%), albeit in a smaller number of cases. For example, DVPs have recently been identified in precursor cervical cancer lesions that are predictive of progression to neoplasia when compared with matched control tissue[21]. The contribution of such DNA methylation outliers in non-cancerous tissues has not yet been evaluated. Further, the distinct functional characteristics of DVPs compared with those of DMPs have not been fully appreciated.

In this study, we determine differential DNA methylation in 52 MZ twin pairs discordant for T1D. In these twin pairs, we perform an EWAS in immune cells known to act as key drivers in the disease process, namely CD4+ T cells, CD19+ B cells and CD14+CD16− monocytes, using Illumina Infinium Human-Methylation450 BeadChips ('450K arrays'). Importantly, our experimental design reduces the impact of all major confounding factors in EWASs, due to the profiling of purified, primary cells derived from MZ twins, who share virtually all somatic variation and early-life environmental exposure[23]. With the exception of one DMP in T cells, we do not identify significant T1D-associated DMPs in any of the investigated immune cell types. However, we find a strong enrichment of DVPs in T1D twins relative to their healthy co-twins. We also observe a cell type-specific enrichment when compared with healthy, unrelated individuals. These

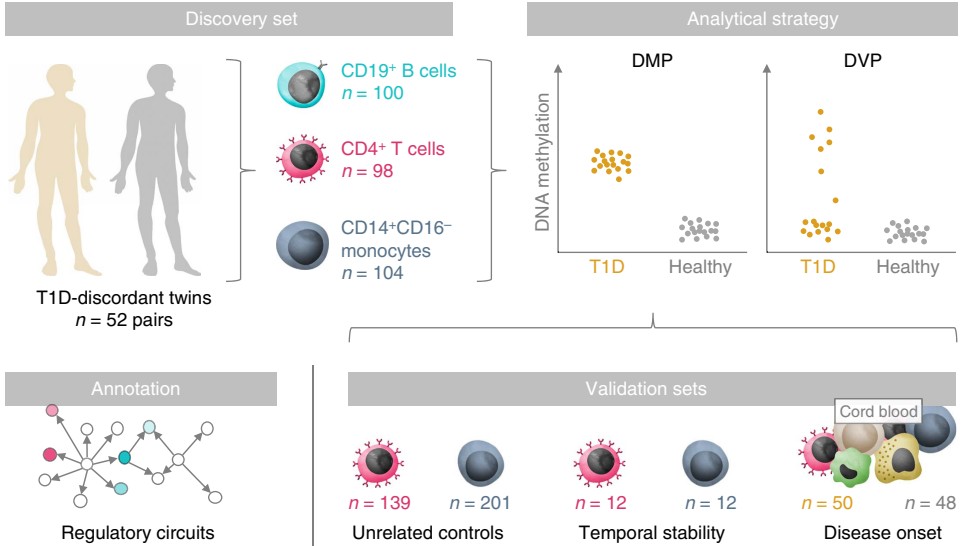

**Figure 1 | Overview of the study design and analytical approach.** We performed an EWAS in 52 MZ twin pairs discordant for T1D in three immune effector cell types: CD4+ T cells, CD19+ B cells and CD14+CD16− monocytes. We used two different approaches to determine differential DNA methylation associated with T1D status in disease-discordant twin pairs. First, we identified DMPs between T1D and healthy co-twins, which correspond to differences in mean DNA methylation levels. Second, we determined DVPs, which reflect heterogeneous 'epigenetic outliers' in T1D twins compared with their healthy co-twins. To assess the biological significance of our findings, we analysed three additional, genome-wide DNA methylation data sets in CD14+ monocytes and CD4+ T cells from 12 T1D-discordant MZ twin pairs; CD14+ and CD4+ cells from 201 and 139 unrelated, healthy individuals; and cord blood from 98 newborns of whom 50 had progressed to overt T1D during childhood. Finally, we characterized T1D-associated DVPs using cell type-specific gene regulatory circuits. Credits: The immune response, Big Picture (http://bigpictureeducation.com/).

T1D-associated DVPs are temporally stable; not under genetic control; enriched at gene regulatory elements; and located at genes involved in immune cell metabolism and the cell cycle.

## Results

**DNA methylation profiles of immune effector cell types**. In 52 T1D-discordant MZ twin pairs, we isolated three immune effector cell types that play a pivotal role in T1D pathobiology: CD4$^+$ T cells, CD19$^+$ B cells and CD14$^+$CD16$^-$ monocytes[1]. Cells were isolated and purified from collected peripheral blood mononuclear cells using magnetic activated cell sorting (MACS). Cell purity of each preparation was evaluated using fluorescence-activated cell sorting (FACS) analysis (Supplementary Fig. 1).

For the discovery stage, we generated a total of 302 genome-wide DNA methylation profiles on the 450K array platform (Fig. 1). The array platform allows the assessment of DNA methylation status at >485,000 CpG sites at single-nucleotide resolution, and covers 99% of RefSeq genes with an average of 17 CpG sites per gene region and 96% of CpG islands[24]. Array data preprocessing and quality control were performed using established analytical tools (see the 'Methods' section), leaving 406,365 CpG sites for subsequent statistical analysis (Supplementary Fig. 2a).

Multidimensional scaling and hierarchical clustering revealed that most of the variation in the data was captured by variation between twin pairs (for example, genetic effects) and cell types (Supplementary Figs 2b and 3). In addition, we performed singular value decomposition to determine principal components of variation in DNA methylation profiles. In our analysis, no principal component was found to correlate with T1D status (Supplementary Fig. 2c).

**Identification of T1D-associated DMPs**. We first measured differences in mean DNA methylation levels between T1D twins and their healthy co-twins in each cell type using a pair-wise analysis. We identified a single DMP at genome-wide significance, cg01674036 in T cells ($P = 2.2 \times 10^{-9}$, false discovery rate (FDR)-corrected $P = 9.1 \times 10^{-4}$, paired $t$ test; Fig. 2a). This DMP demonstrated a mean DNA methylation difference of 2.3% between T1D and healthy co-twins, and mapped to an intergenic region 24.3 kb downstream of the *DDIT4* gene (also known as *REDD1*) encoding DNA-damage-inducible transcript 4 (Fig. 2b). Notably, we did not detect any additional DMPs at an FDR of <0.05 in any of the three cell types.

The DMP cg01674036 co-located with an active gene regulatory region in T cells (Fig. 2c). Chromatin interaction data obtained from a lymphoblastoid cell line provided experimental evidence that this region binds to the promoter region of *DDIT4*. DDIT4 functions as an inhibitor of the mammalian target of rapamycin (mTOR) complex 1; activation of mTOR complex 1 is controlled by anabolic hormones including insulin[25].

The 450K array platform has a fixed set of CpG sites, covering <2% of all annotated CpGs. While this platform is scalable to large sample sizes, the complementary application of sequencing-based approaches is required to comprehensively capture disease-associated DNA methylation loci on a genome-wide level[11,12]. To this end, we further measured DNA methylation levels in CD4$^+$ T cells using whole-genome bisulfite sequencing (WGBS-seq) in four MZ twin pairs, who were originally profiled on the 450K array. In total, we obtained >500 million reads per sample resulting in a mean coverage of between 12.6 and 15.1 reads per CpG site. This allowed us to investigate over 8.7 million CpGs with a minimum coverage of 10 reads across all eight samples (Supplementary Table 1). This analysis was sufficiently powered to detect differentially methylated regions (DMRs) that consist of

at least five CpGs and exhibit a mean DNA methylation difference of >30% at an FDR of <0.05. We did not identify such DMRs to be associated with T1D, irrespective of FDR values.

In conclusion, with the exception of the DMP cg01674036, we did not identify mean DNA methylation differences between T1D twins and their healthy co-twins in any of the three immune cell types using the 450K array platform (Fig. 3a). At genomic loci not covered by the array, results based on WGBS-seq data indicate that mean DNA methylation differences of large effect size are unlikely to exist.

**Identification of T1D-associated DVPs**. Next, we explored whether DNA methylation variability between T1D-discordant MZ twins can shed light on the phenotypic discordance. A recent comparative study[26] demonstrated that current algorithms for DVP detection can substantially differ in terms of their sensitivity and type-1 error rate (see the 'Methods' section). Established algorithms typically assume frequent alterations in the disease phenotype, and thus lack the sensitivity to detect outlier events[26]. Instead, the novel algorithm iEVORA[22], which is based on a regularized version of Bartlett's test, improves the sensitivity to detect DVPs (see the 'Methods' section).

Using iEVORA, we identified 10,548 DVPs in B cells, 4,314 in T cells and 6,508 in monocytes at a stringent FDR of <0.001 (Fig. 3b). Strikingly, in each cell type we found strong enrichment of DVPs that are hypervariable in T1D twins compared with their healthy co-twins ($P < 1 \times 10^{-100}$, binomial test; Fig. 3c). These T1D-associated DVPs represent 'epigenetic outliers' that often occur in individual twin pairs and cell types (Fig. 4a and Supplementary Fig. 4). At DVPs, the DNA methylation differences between the T1D twin and its healthy co-twin were found to be comparatively large in many cases (Supplementary Fig. 4).

We next assessed a range of potential confounding factors that could lead to increased variability in DNA methylation levels, including cellular heterogeneity and differences in cell purification efficiency (as quantified by FACS), age of twins at both disease diagnosis and sample collection, medication use (statins and thyroxine), as well as presence of other autoimmune diseases (thyroiditis, as characterized by thyroid peroxidase autoantibodies). We calculated the fraction of DVPs in T1D twins exhibiting a significant deviation from the healthy co-twins, and then correlated this fraction with different potential confounding variables. For all tested variables, these correlations were not statistically significant ($P > 0.05$; Supplementary Fig. 5).

DNA methylation levels can associate with genetic variants in *cis*, leading to an increase in interindividual DNA methylation variability. Although post-zygotic somatic mutations may occur and give rise to mosaicism in identical twins (with a controlled genetic background)[23], due to the heterogeneous nature of DVPs, we anticipated a modest (if any) genotypic effect on DNA methylation levels at DVPs. To find a definitive answer, we genotyped all 52 twin pairs on Illumina HumanOmni2.5–8 BeadChips, and mapped methylation quantitative trait loci (meQTLs) using a linear-additive modelling approach (see the 'Methods' section). For this analysis, only single-nucleotide polymorphism (SNPs) with a minor allele frequency of >5% and located <50 kb up- and downstream of each CpG site were considered. We found that T1D-associated DVPs (FDR < 0.001) were depleted at meQTLs compared with random sets of CpG sites in all three cell types (permutation $P < 1 \times 10^{-4}$).

In summary, in all three immune cell types we discovered statistically significant DVPs that correlate with T1D status. We provided evidence that these CpG sites are unlikely to be a

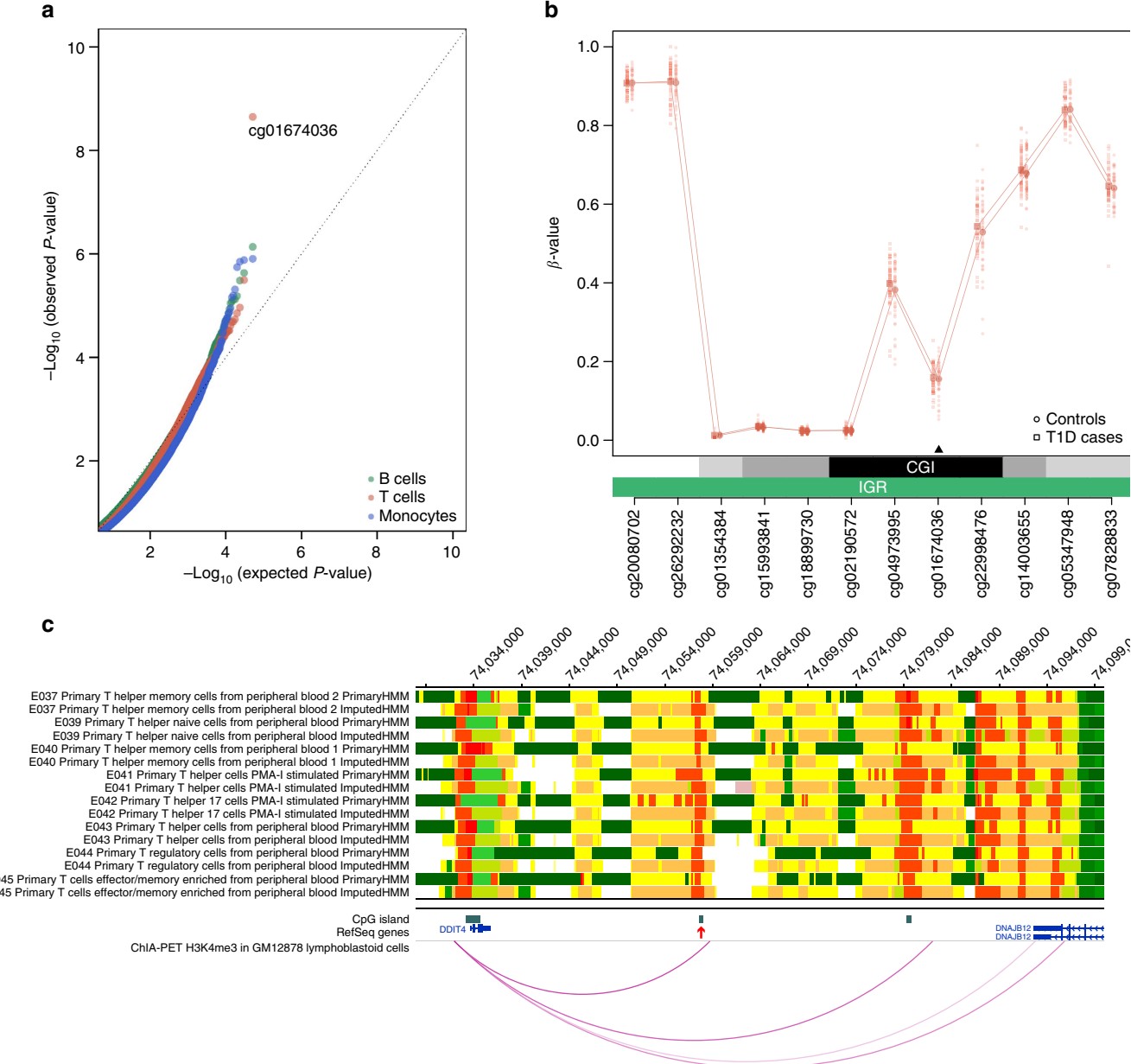

**Figure 2 | Assessment of the functional significance of the T1D-associated DMP cg01674036.** (**a**) QQ plot for the identification of differentially methylated CpG positions (DMPs) between T1D-discordant MZ twin pairs in different immune effector cell types. Only the DMP cg01674036 reached genome-wide significance in T cells, with $P = 2.2 \times 10^{-9}$ (FDR-corrected $P = 9.1 \times 10^{-4}$) and a mean DNA methylation difference of 2.3%. (**b**) Regional plot of the locus harbouring the T-cell-specific DMP cg01674036. The statistically significant DMP is indicated with a black arrow. Data points represent the DNA methylation $\beta$-values (y axis) at the indicated CpGs (x axis) in one individual. For each CpG site, we calculated the mean DNA methylation value (indicated with a larger data point). Every CpG site is annotated with regards to epigenomic feature and gene element using the 450K array annotation manifest. (**c**) Annotation of the genomic locus using epigenomic reference data sets. The genomic locus on chromosome 10q22.1 (position = 74,028,000–74,100,000; genome build = hg19) harbouring the DMP cg01674036 (chr10:74,058,002) is shown using the WashU Epigenome Browser v40.0.0 (http://epigenomegateway.wustl.edu/browser/). The T1D-associated DMP is located at a CpG island (indicated with a red arrow). A total of 16 epigenomic reference tracks provided by the Roadmap Epigenomics project are displayed. Specifically, we show both the primary and imputed chromatin state maps in eight distinct primary T cell populations. The highlighted CpG island overlaps with an active transcription start site (red) or enhancer (orange/yellow) in all available T cell populations. In addition, H3K4me3 ChIA-PET data in the lymphoblastoid cell line GM12878 revealed a long-range chromatin interaction between the active regulatory element and the gene promoter region of *DDIT4*. CGI, CpG island; ChIA-PET, chromatin interaction analysis by paired-end tag sequencing; IGR, intergenic region.

consequence of confounding factors, including cellular heterogeneity, and that they act independently of genetic variation.

**Temporal stability of T1D-associated DVPs.** Following the discovery of DVPs on 450K arrays, we reassessed the

T1D-associated DNA methylation hypervariability phenotype in twins after five years, using a second assay platform. We retrieved genome-wide DNA methylation profiles of CD14$^+$ monocytes and CD4$^+$ T cells from 12 T1D-discordant MZ twin pairs generated on Illumina Infinium HumanMethylation27 BeadChips

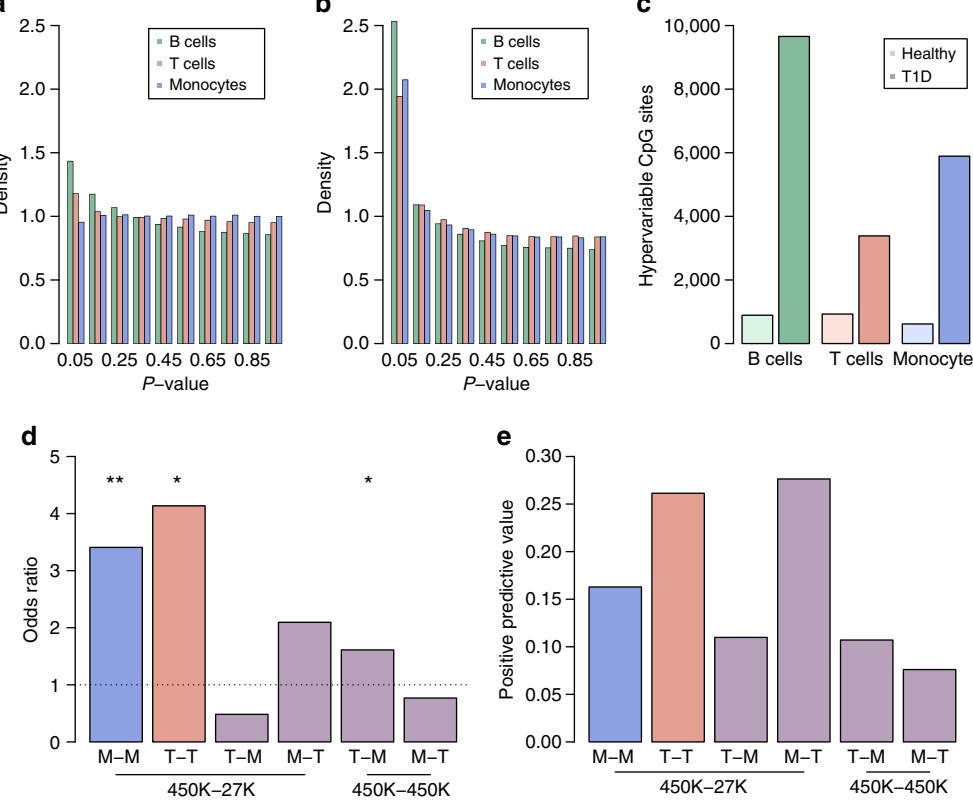

**Figure 3 | DNA methylation variation in identical twin pairs discordant for T1D. (a)** Histogram of *P*-values for the identification of differentially methylated CpG positions (DMPs) between T1D-discordant MZ twin pairs in different immune effector cell types. DMPs were determined using a paired *t* test. **(b)** Histogram of *P*-values for the identification of T1D-associated differentially variable CpG positions (DVPs). DVPs were determined at an FDR of <0.001 using the algorithm iEVORA. **(c)** Bar plots showing the enrichment of DVPs in T1D twins compared with their healthy co-twins. While this hypervariability phenotype was found in all cell types ($P < 1 \times 10^{-100}$, binomial test), it was particularly pronounced in B cells. **(d)** Bar plots showing the odds ratios of the assessment of temporal stability of T1D-associated DVPs in an external data set of CD14$^+$ and CD4$^+$ cells derived from 12 disease-discordant MZ twin pairs generated on 27K arrays. Importantly, the identified DVPs in CD14$^+$ and CD4$^+$ cells replicated in a cell type-specific context. Stars denote statistical significance assessed using a one-tailed Fisher's exact test: *$P < 1 \times 10^{-2}$ and **$P < 1 \times 10^{-4}$. **(e)** Positive predictive values for the analyses shown in **d**. B, CD19$^+$ B cells; M, CD14$^+$CD16$^-$ monocytes; T, CD4$^+$ T cells.

('27K arrays')[15]. These twins belonged to the same twin registry used for the discovery cohort of this study, but provided new DNA samples for reassessment after five years.

We confirmed directionality of the T1D-associated DVPs (FDR < 0.001), indicating robust technical detection across assay platforms in both CD14$^+$ cells ($P = 7.7 \times 10^{-5}$, one-tailed Fisher's exact test; Fig. 3d) and CD4$^+$ cells ($P = 7.8 \times 10^{-3}$; Fig. 3d). Consistently, the attained positive predictive values were higher in the direct cell type comparison (Fig. 3e).

Taken together, we showed that DNA methylation levels at T1D-associated DVPs are temporally stable over at least five years in patients with established diabetes, and can be observed across two assay platforms.

**Evaluation of T1D-associated DVPs in unrelated individuals.** We further assessed T1D-associated DVPs using independent genome-wide DNA methylation profiles retrieved from the BLUEPRINT Consortium. Specifically, we reasoned that DVPs hypervariable in T1D ought to be hypervariable when compared with an external set of healthy controls. We obtained 450K array data sets of CD14$^+$ and CD4$^+$ cells derived from 201 and 139 unrelated, healthy individuals, respectively. These individuals were drawn from a population of blood donors, and thus are unlikely to have strong genetic susceptibility to T1D.

DVPs that were found to be hypervariable in T1D twins compared with their healthy co-twins, were also hypervariable when compared with unrelated individuals with limited genetic susceptibility markers (Supplementary Fig. 6). As demonstrated before, DVPs showed cell type specificity ($P = 1.3 \times 10^{-60}$ and $P = 4.5 \times 10^{-107}$, for monocytes and T cells, respectively).

In conclusion, our analysis provided further evidence that the identified DVPs represent relevant, cell type-specific markers for T1D.

**Assessment of T1D-associated DVPs in cord blood.** To explore whether the identified DVPs emerged before the onset of T1D, we generated genome-wide DNA methylation profiles of umbilical cord blood obtained from newborns. These newborns were part of the DiPiS cohort, a population-based prospective study of T1D in children[27]. We selected samples from 98 newborns of whom 50 had progressed to overt T1D during childhood, while 48 did not. We hypothesized that if the T1D-associated DVPs (that are independent of genetic risk factors) were already observed in cord blood before disease onset, they could potentially contribute to T1D pathogenesis or be an early indicator of disease.

We correlated DNA methylation levels at T1D-associated DVPs identified in purified immune cell types with those in cord blood tissue. This assessment did not reach statistical significance ($P > 0.05$, one-tailed Fisher's exact test).

Based on these findings, we conclude that the discovered DVPs occur post-birth and are likely associated with the pathogenesis of T1D either before or after the clinical diagnosis. Nonetheless,

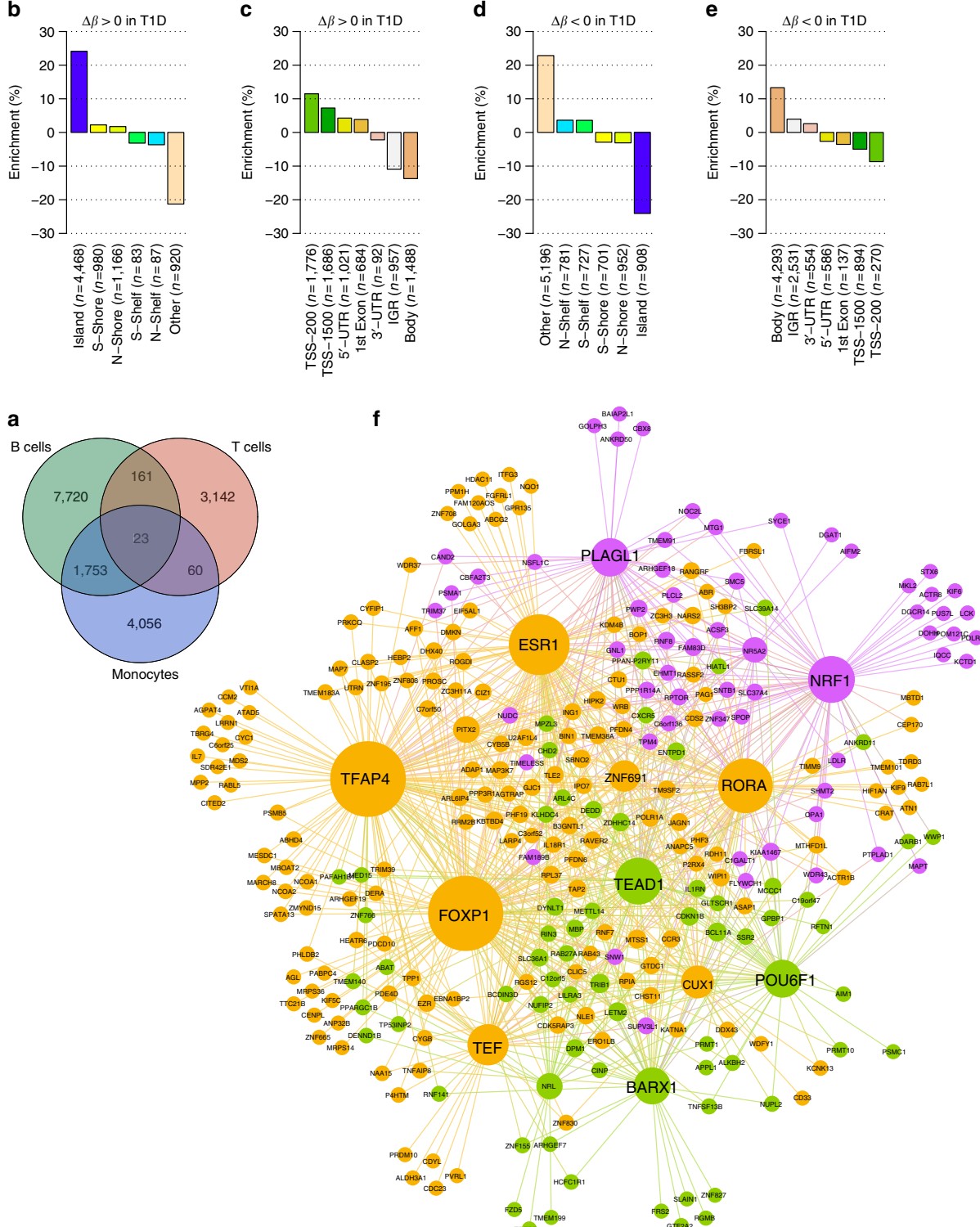

**Figure 4 | Functional annotation of T1D-associated DVPs.** (**a**) Venn diagram showing the overlap of T1D-associated DVPs (FDR < 0.001) across cell types. Although many of the identified DVPs were found to be cell type-specific, B cells and monocytes showed a substantial proportion of overlap. (**b,c**) Enrichment of T1D-associated DVPs at different epigenomic features and gene elements. Here, only DVPs at which the DNA methylation level was increased (hypermethylated; $\Delta\beta > 0$) in T1D twins compared with their healthy co-twins are shown. The enrichment is shown in relation to all 450K array probes that passed quality control. (**d,e**) The same analyses as shown in **b** and **c**, but for DVPs at which the DNA methylation level was reduced (hypomethylated; $\Delta\beta < 0$) in T1D twins. (**f**) Integration of T1D-associated DVPs with gene regulatory circuits in CD19$^+$ B cells. The network was constructed using the corresponding genes of all T1D-associated hypomethylated DVPs that map to gene promoters and hypermethylated DVPs at gene bodies identified in B cells. The resulting network consisted of 297 genes connected via 906 regulatory edges. Three network modules were identified and are highlighted in different colours: Module 1 (*n* = 61 genes) is shown in purple, module 2 (*n* = 69) in green and module 3 (*n* = 167) in orange. These modules were further characterized using functional enrichment analysis (Supplementary Table 2 and Supplementary Table 3). IGR, intergenic region; N, north, that is, upstream; S, south, that is, downstream; TSS200/1500, 200/1500 bp upstream of a transcription start site; UTR, untranslated region.

it should be noted that cord blood, similar to peripheral whole blood, is a substantially heterogeneous tissue that hampers the precise measurement of DNA methylation levels. Our study is by far the most powered in this area, but future studies with increased statistical power will be needed to corroborate our conclusions.

**Functional significance of T1D-associated DVPs**. Next, we investigated whether T1D-associated DVPs exhibit a certain level of functional organization and whether the nearby genes cluster in biological pathways. First, we performed enrichment analyses with regards to gene elements and epigenomic features as defined in the 450K array annotation manifest. We distinguished between T1D-associated DVPs (FDR $< 0.001$) at which the DNA methylation level is either increased (hypermethylated) or decreased (hypomethylated) in T1D twins compared with their healthy co-twins. This discrimination is important, because the effect of DNA methylation on the regulation of gene expression is distinct at different gene elements[5,28]. Across the three immune cell types, we found an enrichment of T1D-associated hypermethylated DVPs at CpG islands ($P = 1.5 \times 10^{-265}$, hypergeometric test; Fig. 4b) and proximal gene promoters ($P = 2.3 \times 10^{-131}$; Fig. 4c). In contrast, hypomethylated DVPs were depleted at CpG islands ($P < 2.2 \times 10^{-308}$; Fig. 4d) and enriched at gene bodies ($P = 1.0 \times 10^{-47}$; Fig. 4e).

We corroborated these enrichment patterns using cell type-specific chromatin state maps. We retrieved reference chromatin state data based on five chromatin marks in primary cells from peripheral blood, provided by the Roadmap Epigenomics project[29] (see the 'Methods' section). For all three immune cell types, we found cell type-specific enrichment of T1D-associated hypermethylated DVPs at chromatin states marking active transcription start sites proximal to gene promoters ($P < 2.2 \times 10^{-308}$ in B cells, $P = 4.2 \times 10^{-38}$ in T cells, and $P = 5.2 \times 10^{-262}$ in monocytes, hypergeometric tests; Supplementary Fig. 7a). We also observed depletion at states marking enhancers ($P = 6.2 \times 10^{-269}$ in B cells, $P = 3.4 \times 10^{-22}$ in T cells, and $P = 1.3 \times 10^{-258}$ in monocytes; Supplementary Fig. 7a). Hypomethylated DVPs showed inverted enrichment patterns (Supplementary Fig. 7b).

We then tested for enrichment of biological process ontology terms attributed to genes in proximity to T1D-associated DVPs. We adjusted for the differing number of CpGs per gene present on the 450K array to reduce bias in the gene set analysis[30]. Across all cell types, the T1D-associated DVPs cumulatively clustered at genes involved in molecular metabolic processes and the cell cycle (Supplementary Fig. 8). The enrichment was particularly pronounced in B cells (Supplementary Fig. 8).

In summary, these findings showed that T1D-associated DVPs localize at specific gene regions and active chromatin states implicated in the regulation of gene expression, and highlighted gene pathways related to cell metabolism and the cell cycle.

**Integration of T1D-associated DVPs with regulatory circuits**. Finally, using a gene regulatory network approach we further explored the T1D-associated DVPs in B cells that may lead to transcriptional regulation of relevant pathways. We obtained CD19$^+$ B-cell-specific regulatory circuits[31] that consist of interactions between transcription factors and genes derived from genome-wide promoter and enhancer activity maps presented by the FANTOM5 project[32,33]. We defined two sets of DVPs that may lead either to transcriptional repression or activation in B cells, and assigned these to their corresponding genes (see the 'Methods' section). Then, we intersected the resulting gene lists with the regulatory circuits.

The regulatory network created using 'gene-repressing' DVPs consisted of 1,465 genes and 16,712 regulatory edges. The corresponding network constructed using 'gene-activating' DVPs consisted of 297 genes connected via 906 edges. While we were unable to resolve the structure of the gene-repressing network and could not identify significant modules within this network, the gene-activating network showed three network modules (Fig. 4f). We further characterized these modules using gene enrichment analyses (see the 'Methods' section), and the results of all gene set analyses are shown in detail in Supplementary Table 2.

Module 1 contained 61 genes, including *NRF1* encoding nuclear respiratory factor 1 (Fig. 4f; shown in purple). NRF1 is a transcription factor that regulates the expression of genes encoding key enzymes in oxidative metabolism and mitochondrial function[34]. The module showed enrichment in ontology terms related to glucose-6-phosphate transmembrane transporter activity. Further analysis revealed overrepresentation of genes involved in mTOR signalling, a central pathway in the regulation of cell metabolism, growth and proliferation (Supplementary Table 3)[35]. Module 2 contained 69 genes (Fig. 4f; shown in green), and was enriched for genes connected to interleukin-1 receptor binding and receptor antagonist activity. This pathway is implicated in T1D-associated altered innate immunity[36]. Module 3 contained 167 genes, including the *FOXP1* gene hub (Fig. 4f; shown in orange). *FOXP1* encodes forkhead box P1, an important transcriptional regulator of B cell, T cell and monocyte differentiation. Recent studies in mice also demonstrated that Foxp1 is essential for islet α-cell proliferation and function[37], and plays a key role in the regulation of systemic glucose homeostasis[38].

The integration of T1D-associated DVPs with gene regulatory circuits in CD19$^+$ B cells confirmed our initial findings (Supplementary Fig. 8), and further implicated signalling pathways related to immune cell metabolism. While alteration in these pathways could be secondary to the systemic metabolic abnormalities associated with diabetes, we note that these pathways could also predispose to autoimmune diseases including T1D.

**Discussion**

In this study, we investigated whether differential epigenetic variation can explain discordance of T1D in identical twins. We measured genome-wide DNA methylation levels in 52 twin pairs across disease-relevant immune effector cell types. Our unique study design allowed us to reduce confounding factors that have impeded many previous EWASs, namely cellular heterogeneity (by using multiple, sorted, primary cell types) and genetic heterogeneity, age and early-life environmental effects (by using disease-discordant MZ twins).

Notably, with the exception of a single T-cell-specific DMP, we did not detect convincing differences in mean DNA methylation associated with T1D in our MZ twin cohort using the 450K array platform (FDR $< 0.05$; Fig. 2a). The DMP that did reach statistical significance, cg01674036 in CD4$^+$ T cells, is not contained on the 27K array platform and therefore could not be technically replicated in our data set. Annotation using epigenomic reference data sets revealed that the DMP maps to an active gene regulatory region in T cells and interacts with the gene promoter of *DDIT4* (Fig. 2c). The corresponding protein is involved in the mTOR signalling pathway, which has been implicated in the gene network analysis of T1D-associated DVPs (Fig. 4f and Supplementary Table 3).

In addition, we did not find DMRs of large effect size ($> 30\%$; $\geq 5$ CpGs) in four disease-discordant MZ twin pairs

using WGBS-seq. It is possible that T1D-associated DMPs and DMRs could be discovered in much larger cohorts or more highly selected cell populations using either Infinium arrays or bisulfite sequencing[39]. In particular, the recent availability of the Illumina Infinium MethylationEPIC BeadChip covering over 850,000 CpG sites[40], of which many are located at enhancer regions identified by the ENCODE and FANTOM5 projects, may allow for discovery of additional T1D-associated loci. However, if they were present, such loci would most likely be of small effect size. Indeed, this notion would be consistent with findings from genome-wide association studies (GWASs) of T1D and other complex traits and diseases. In GWASs, many hundreds of trait-associated genetic variants have been identified, the vast majority of which possess small effect sizes[41].

In contrast, we identified a substantial number of CpGs that are hypervariable in T1D twins compared with their healthy co-twins. The DNA methylation differences at DVPs were found to be comparatively large in many cases (Supplementary Fig. 4). The sensitive yet robust identification of DVPs is challenging, and is characterized by a high type-1 error rate[22]. Thus, replication of our findings in independent sample cohorts in future studies is paramount.

Here, for the first time, we detected and functionally annotated DVPs in a common disease phenotype other than cancer. DVPs have been shown to correlate with the early stages of carcinogenesis. Consistently, our data suggest that T1D-associated DVPs are associated with T1D after clinical diagnosis. In this regard, causal inference analysis may be applied to further characterize and quantify the extent of the relationship between genetic variants, epigenetic variants and phenotypic discordance[14,42]. Future longitudinal studies of pre-diabetic individuals will establish whether the epigenetic changes antedate the clinical diagnosis.

Our findings have important implications for the future application of the EWAS approach to elucidate human disease mechanisms. First, the use of purified, primary cell populations likely reduced the overall number of association signals typically detected in case-control EWASs conducted in peripheral blood, but with the critical advantage of yielding genuine disease-relevant signals, if present. Second, we propose the complementary assessment of DNA methylation variability in parallel to mean DNA methylation for any future EWAS. Analytical tools are now readily available to identify DVPs for other complex traits and diseases[19,22].

Our results showed that DVPs differ considerably between the profiled cell types (Fig. 4a and Supplementary Fig. 4), suggesting that the response of each cell type is specific. The importance of the three immune effector cell types used in this study in the development of T1D has been recognized through experimental evidence[1,3]. Furthermore, genetic variants associated with T1D are enriched at enhancer sequences active in T and B cells, as well as CD34$^+$ stem cells and thymus tissue[43]. However, it is plausible that other cell types not assayed here may contain relevant DNA methylation differences. Alternatively, it may also be possible that rare sub-populations of the three immune cell types, such as regulatory T cells (CD25$^+$FOXP3$^+$ cells) or T helper 17 cells (T$_H$17 cells), harbour epigenetic signals that remain undetected in the broad population of CD4$^+$ T cells[1,44,45]. Future EWASs in subsets of T cells may be conducted to address this possibility.

Of note, a recent report found an increase in DNA hydroxymethylation levels at gene promoters in CD4$^+$ T cells in patients with systemic lupus erythematosus, an autoimmune disease, compared with healthy controls[46]. DNA hydroxy-methylation remodelling has also been observed in CD4$^+$ T cell differentiation[47]. As our experimental approach did not allow the discrimination between methylated and hydroxymethylated cytosine bases, DNA hydroxymethylation could thus contribute to the observed differential variability, potentially providing a general mechanism underlying the pathogenesis of autoimmune diseases.

Studies have reported the co-localization of meQTL at genetic risk loci of complex traits and common diseases identified through GWASs, including schizophrenia[13], blood pressure[10], and several cancer types[48]. Consequently, we overlapped our T1D-associated DVPs with 59 T1D genetic susceptibility loci retrieved from T1DBase, a curated web resource (http://www.t1dbase.org; v4.19). We did not find a statistically significant enrichment of DVPs at these loci ($P > 0.05$, hypergeometric test). A specific enrichment test of T1D-associated DVPs mapping to the major histocompatibility complex (MHC) locus also did not achieve statistical significance compared with all assessed CpG sites (Supplementary Fig. 9). The MHC locus is key in conferring genetic risk of T1D and other autoimmune diseases, as it harbours many genes encoding cell surface molecules that orchestrate components of the immune system. This analysis provided further evidence that T1D-associated genetic and epigenetic variants appear to act independently.

We have identified T1D-related DVPs in immune effector cells that associate with genes involved in cell metabolism and the cell cycle (Supplementary Fig. 8 and Supplementary Table 2). Specifically, by integrating T1D-associated DVPs with gene regulatory circuits in CD19$^+$ B cells, we pinpointed key transcriptional regulators such as NRF1 and FOXP1 (Fig. 4f), and pathways such as mTOR signalling (Supplementary Table 3). Indeed, the same signalling pathways have been implicated in differentiation, proliferation and metabolism of both T cells and monocytes[49–52]. For example, deletion of *Foxp1* in naïve CD8$^+$ T cells leads to activation of the mTOR signalling cascade[53], indicating a relationship between gene modules 1 and 3 of the regulatory network we identified here (Fig. 4f). Therefore, DVPs could modulate disease activity through the regulation of immune effector cell gene expression either before or after the induction of the disease process.

However, it remains possible that other T1D-associated DVPs result from disease-associated metabolic disturbances. Previous studies reported DMPs (including cg19693031) at the *TXNIP* gene to be inversely correlated with both type 2 diabetes and sustained hyperglycaemia (for example, haemoglobin A1c levels)[54,55]. In our data set of T1D patients, we also found cg19693031 to be a DVP in monocytes ($P = 9.1 \times 10^{-4}$); this observation suggests that a proportion of DVPs result from the diabetes-associated metabolic effect. In either case, it is likely that the impact of epigenetic changes on T1D-associated immune effector cells would adversely affect the natural history of the disease[3].

The exact mechanism by which epigenetic instability in T1D is manifested, its timing in relation to induction of islet auto-immunity, as well as its impact on disease progression, remains to be explored. However, these questions can now be addressed through the study of individuals at high T1D-risk and those with variable disease severity. In this way, we can achieve our ultimate aim of identifying diagnostic and prognostic epigenetic biomarkers that can improve the management of T1D.

## Methods

**Ethics statement.** This study was approved by the Northern and Yorkshire Research Ethics Committee (REC reference number: 06/MRE03/22) and the NRES Committee East of England-Hertfordshire (12/EE/0040). All participants gave informed consent either personally or by parental consent, as appropriate.

**Study samples.** MZ twin pairs were ascertained by referral through their physicians to the British Diabetic Twin Study, the Barbara Davis Center for

Childhood Diabetes and Diabetes Prevention TrialNet (USA) and the BMBF Pediatric Diabetes Biobank (Germany). T1D-associated autoantibodies were analysed by radioimmunoassay[56,57]. We established monozygosity by means of DNA fingerprinting using an AmpFLSTR Identifiler PCR Amplification Kit (Life Technologies) and consultation of clinical data. T1D status was established by standard criteria[58]. T1D patients have been treated from diagnosis with insulin and take highly purified human insulin at least twice daily. We excluded twins who were pregnant and twins with significant co-morbidities including severe macrovascular and microvascular complications of diabetes. Umbilical cord blood was obtained from 35,000 newborns enroled in the DiPiS Study (Sweden)[27] between the years 2000 and 2004. From this cohort, we selected 98 neonates of whom 50 progressed to T1D and 48 did not. Children were followed for 15 years to monitor if they develop markers of islet autoimmunity and T1D. The samples consisted of dried cord blood spots dotted onto cards.

**Cell sorting and purity analysis.** Peripheral blood mononuclear cells were prepared from 50 ml of heparinized blood using Percoll density gradient separation. CD4$^+$ T cells, CD19$^+$ B cells and CD14$^+$CD16$^-$ monocytes were isolated using MACS according to the manufacturer's instruction. First, CD19$^+$ B cells were separated with CD19 MicroBeads (130-050-301, Miltenyi Biotec). The negative fraction was then washed and incubated with CD16 MicroBeads (130-045-701, Miltenyi Biotech). The fraction depleted of CD16$^+$ cells was selected for CD14$^+$ monocytes using CD14 MicroBeads (130-050-201, Miltenyi Biotech). Finally, the resulting negative fraction was further incubated with CD4 MicroBeads (130-045-101, Miltenyi Biotech) to obtain CD4$^+$ T cells. Based on the number of isolated peripheral blood mononuclear cells, we used 50 µl of CD19 MicroBeads, 20 µl of CD14 MicroBeads, and 20 µl of CD4 MicroBeads per 10 million total cells. We assessed the purified cell populations with FACS. The following antibodies were used at a dilution of 1:11 per 10 million total cells for each cell type: 20 µl of FITC-conjugated mouse anti-human CD14 clone MφP9 (345784, BD Biosciences) and 10 µl of CD4 clone M-T466 (130-080-501, Miltenyi Biotech); 10 µl of phycoerythrin (PE)-conjugated mouse anti-human CD19 clone LT19 (130-091-247, Miltenyi Biotech) and 20 µl of CD16 clone B73.1/leu11c (332779, BD Biosciences); 5 µl of PerCP-Cy5.5-conjugated mouse anti-human CD64 clone 10.1 (561194, BD Biosciences); and 5 µl of PE-Cy7-conjugated mouse anti-human CD45 clone HI30 (MHCD4512, Invitrogen). Cells were incubated with antibodies at 4 °C for 15 min, washed with 2 ml of phosphate-buffered saline (PBS) and ethylenediaminetetraacetic acid (EDTA), and re-suspended in a volume of 500 µl for FACS analysis. Across all cell types, the mean cell purity was 90%.

**DNA extraction.** We extracted genomic DNA from MACS-enriched cell populations using a QIAamp DNA Blood Mini Kit (QIAGEN) according to manufacturer's instructions. DNA was extracted from cord blood using a GenSolve DNA Recovery Kit (Labtech) according to the manufacturer's instructions. DNA concentration was determined using a Qubit dsDNA HS Assay Kit (Invitrogen) and DNA integrity visually inspected on a 2% agarose gel.

**Illumina Infinium HumanMethylation450 assay.** Genomic DNA was bisulfite-converted using an EZ-96 DNA Methylation MagPrep Kit (Zymo Research) according to the manufacturer's instructions. We applied 500 ng of genomic DNA to bisulfite treatment, and eluted purified, bisulfite-converted DNA in 20 µl of M-Elution Buffer (Zymo Research). DNA methylation levels were measured on Infinium HumanMethylation450 BeadChips (Illumina) following the manufacturer's protocol. In brief, 4 µl of bisulfite-converted DNA was isothermally amplified, enzymatically fragmented and precipitated. Next, precipitated DNA was resuspended in hybridization buffer and dispensed onto the BeadChips. To limit batch effects, samples were randomly distributed across slides and arrays. The hybridization was performed at 48 °C for 20 h using a Hybridization Oven (Illumina). After hybridization, BeadChips were washed and processed through a single-nucleotide extension followed by immunohistochemistry staining using a Freedom EVO robot (Tecan). Finally, the BeadChips were imaged using an iScan Microarray Scanner (Illumina).

**Illumina Infinium HumanMethylation450 data preprocessing.** The DNA methylation fraction at a specific CpG site was calculated as $\beta = M (M + U + 100)^{-1}$, for which M and U denote methylated and unmethylated fluorescent signal intensities, respectively. The $\beta$-value statistic ranges from absent ($\beta = 0$) to complete DNA methylation ($\beta = 1$) at a specific CpG. We normalized the 450K array data using BMIQ (Beta MIxture Quantile dilation), an intra-array normalization method that adjusts the $\beta$-values of type-2 design probes into a statistical distribution characteristic of type-1 probes[59]. Next, we filtered (1) probes with median detection $P$-value $\geq 0.01$ in one or more samples; (2) probes with bead count of <3 in at least 5% of samples; (3) probes mapping to sex chromosomes; (4) non-CG probes; (5) probes mapping to ambiguous genomic locations[60]; and (6) probes harbouring annotated SNPs within 2 bp of the probed CG irrespective of allele frequency in the European populations, as reported by dbSNP v135 (ref. 60). Finally, we adjusted for known batch effects using an empirical Bayesian framework[61], as implemented in the ComBat function of the R package SVA[62]. The final data matrix consisted of $\beta$-values across 406,365 CpG

sites × 302 samples, that is, 49, 50 and 52 MZ twin pairs in T cells, B cells and monocytes, respectively.

**Identification of DMPs and DVPs.** To identify DMPs, we applied a paired $t$ test and estimated the FDR using the R package q-value[63]. DVPs were identified using iEVORA[22], an algorithm based on a regularized version of Bartlett's test. The algorithm is freely available as an executable R script from the Supplementary Information of the publication at http://www.nature.com/ncomms/. A disadvantage of Bartlett's test is that single outliers can drive the DVP ranking. Therefore, iEVORA uses a novel procedure to regularize Bartlett's test, by selecting CpGs based on significant Bartlett's test $P$-values, but ranking these selected features according to $t$ test $P$-values[22]. This heuristic method guarantees (1) that selected CpGs are significant DVPs; and (2) that the ranking favours DVPs that are either DMPs at genome-wide significance or as close to being DMPs as possible. This regularization step favours DVPs that are driven by more frequent outliers compared with DVPs driven by single outliers. Bartlett's test $P$-values from iEVORA are corrected for multiple testing using the FDR method implemented in the R package q-value. To keep the number of false positives as small as possible, avoiding any impact on the top-ranked features, a stringent FDR of <0.001 was used. Of note, application of an alternative approach, DiffVar (ref. 64), did not reveal significant DVPs at an FDR of <0.05. DiffVar compares the absolute deviations from the respective group means using a (moderated) $t$ test, as the method assumes that the differential variability is driven by numerous outliers within a disease phenotype. This algorithm offers improved control of the type-1 error rate at the expense of reduced power[26]. Thus, iEVORA can be seen as a compromise between DiffVar (which ignores differential variability driven by few outliers resulting in a low type-1 error rate and low sensitivity) and EVORA[19] (which favours differential variability driven by single outliers resulting in much greater sensitivity albeit at the expense of a higher type-1 error rate).

**WGBS-seq data preprocessing and DMR calling.** Sample preparation and preprocessing of WGBS-seq data were conducted using previously established protocols and pipelines[65]. Sequencing statistics are provided in Supplementary Table 1. Counts of unmethylated and methylated cytosine in the context of CpG sites were extracted from the mapped BAM files using a publicly available algorithm (https://bitbucket.org/lowelabqmul/bs-seq-dmr-caller). In brief, the algorithm uses a windowslope approach that progresses along the genome and determines groups of CpG sites that have the same directional difference between cases and controls. The method requires each CpG to be located within 1,000 bp of its neighbouring CpG. To determine the significance of the DMR, the $\chi^2$ statistic is calculated for the pooled counts across each of the CpGs at the locus and across all the samples. The sample identities are then permuted and a new statistic is calculated; this is repeated 1,000 times, and the original statistic is compared with the permutated statistics to produce a $P$-value for each CpG. Then, the $P$-value for each CpG is combined into a single $P$-value using Fisher's method. Finally, the FDR is estimated for each of the DMRs using the R package q-value[63].

**Whole-genome genotyping and meQTL mapping.** The quantity and integrity of DNA samples were assessed using a NanoDrop spectrophotometer (Thermo Scientific). Samples were normalized to a concentration of 50 ng µl$^{-1}$ before amplification. Then, DNA was hybridized to Infinium HumanOmni2.5–8 v1.2 BeadChips (Illumina), according to the manufacturer's instructions. Following genotyping, raw data were imported into GenomeStudio (Illumina), and genotypes called using the standard cluster file provided by the arrays. Quality checks, including comparisons with called versus reported sex and genotype consistency between twins, were performed using GenomeStudio. We excluded all SNPs with a minor allele frequency of <5% and Hardy–Weinberg equilibrium $<1 \times 10^{-6}$, leaving 609,587 SNPs for subsequent meQTL analysis. Further, we confirmed matching DNA methylation and genotype data sets by comparison of genotype calls across the Infinium platforms. To investigate whether DNA methylation levels at DVPs are correlated with genotypes, we mapped meQTLs genome-wide using the software Matrix eQTL[66]. We applied standard parameters except the $P$-value output threshold was set to $1 \times 10^{-8}$ and the maximum distance between interactions of CpGs and SNPs was set to 100,000 bp. We included the following covariates in the linear-additive model: age, sex, batch and T1D status. The analysis identified 13,579 CpG sites for T cells, 11,790 for B cells and 15,531 for monocytes that correlated with at least one SNP. Then, we determined whether T1D-associated DVPs are enriched at meQTLs compared with random sets of CpGs ($n = 10,000$).

**Assessment of DVPs in additional data sets.** We retrieved DNA methylation profiles of CD14$^+$ monocytes and CD4$^+$ T cells from 12 T1D-discordant MZ twin pairs generated using 27K arrays[15]. In addition, we used 450K array DNA methylation profiles of CD14$^+$ and CD4$^+$ cells from 201 and 139 unrelated, healthy individuals, respectively, obtained from the BLUEPRINT Consortium. From the DVPs identified using the 450K array in the discovery stage (FDR<0.001), we selected all probes that were also present in the external data set. First, we computed the log-ratio of the variances in T1D twins versus healthy co-twins. To assess congruence between the discovery and validation sets, we then

calculated the log-ratio of the variances in each set against each other. Finally, we counted the number of selected DVPs with significant $P$-values in the external set and the subset of those that were hypervariable and hypovariable in T1D cases. This resulted in a $2 \times 2$ table, with a subsequent Fisher's exact test allowing us to statistically assess whether the selected DVPs validate in the external set.

**Functional annotation of T1D-associated DVPs.** For the enrichment analyses with regards to gene elements and epigenomic features, we used the annotation provided by the 450K array annotation manifest. For the analyses with regards to chromatin states, we retrieved data generated using the core 15-state ChromHMM model based on five chromatin marks (H3K4me3, H3K4me1, H3K36me3, H3K27me3 and H3K9me3) from http://egg2.wustl.edu/roadmap/web_portal/. We selected chromatin states maps of primary B cells, T cells and monocytes from peripheral blood, corresponding to the reference epigenome identifiers E032, E034 and E029, respectively. Enrichment was assessed by repeated random sampling ($n = 1,000$) using all probes that passed quality control. T1D-associated DVPs were linked to genes using the 450K array annotation manifest[24]. Then, by applying the function gometh implemented in the R package missMethyl[30], genes were associated with ontology terms and enrichment of these terms was calculated in relation to all CpG sites on the 450K array platform that passed quality control. This method takes account of the differing number of probes per gene present on the 450K array.

**Analysis of CD19$^+$ B-cell-specific regulatory circuits.** We retrieved the CD19$^+$ B cell regulatory network from http://regulatorycircuits.org. The network consists of 11,997 nodes (genes) and 1,148,319 edges (interactions between transcription factors and regulatory elements of target genes). We selected all T1D-associated DVPs that were genome-wide significant ($P = 1.2 \times 10^{-7}$). 'Gene-activating' DVPs were defined as CpG sites that were either hypomethylated in T1D twins compared with their healthy co-twins and annotated as TSS1500, TSS200, 5′-UTR or 1stExon on the 450K array annotation manifest; or hypermethylated and annotated as Body or 3′-UTR. Accordingly, 'gene-repressing' DVPs were defined as CpGs that showed hypermethylation at gene promoters or hypomethylation at gene bodies. We only considered genes that directly interacted with other genes of the defined gene set. Network modules were identified using Gephi (http://gephi.org) and the Lovain method[67]. Then, we performed functional enrichment analyses of these modules using the R packages GOstats[68] and ReactomePA[69]. We tested for overrepresentation of gene ontology (GO) molecular function terms using the following parameters: conditional = TRUE and FDR < 0.25 (Benjamini and Hochberg method[70]). We performed further functional enrichment tests of the network modules at an FDR of < 0.01 using Cytoscape[71] and ClueGO[72]. For these tests, we specified the following ontologies: GO Biological Process, GO Immune System Process, GO Molecular Function, KEGG, REACTOME and WikiPathways. We applied GO Term Fusion and a minimum number of three genes or 4% of all genes for the corresponding GO category or pathway. The kappa score was set to 0.4. All enrichment analyses of network modules were contrasted to all genes in the whole regulatory network that were also associated with CpG sites passing quality control on the 450K array platform ($n = 10,660$).

**Data availability.** All 450K array and WGBS-seq data sets that support the findings of this study have been deposited in the European Genome-phenome Archive (EGA) with the accession code EGAS00001001598 (https://www.ebi.ac.uk/ega/studies/EGAS00001001598). We retrieved 450K array data sets of CD14$^+$ and CD4$^+$ cells from EGA with the accession code EGAS00001001456 (https://www.ebi.ac.uk/ega/studies/EGAS00001001456), and 27K array data sets of CD14$^+$ and CD4$^+$ cells from Gene Expression Omnibus (GEO) with the accession code GSE56606 (http://www.ncbi.nlm.nih.gov/geo/query/acc.cgi?acc=GSE56606).

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

## Acknowledgements

This work was funded by the EU-FP7 project BLUEPRINT (282510) and the Wellcome Trust (99148). We thank all twins for taking part in this study; Kerra Pearce and Mark Kristiansen (UCL Genomics) for processing the Illumina Infinium HumanMethylation450 BeadChips; Rasmus Bennet for technical assistance; and Laura Phipps for proofreading the manuscript. The BMBF Pediatric Diabetes Biobank recruits patients from the National Diabetes Patient Documentation System (DPV) and is financed by the German Ministry of Education and Research within the German Competence Net Diabetes Mellitus (01GI1106 and 01GI1109B). It was integrated into the German Center for Diabetes Research in January 2015. We thank the Swedish Research Council and SUS Funds for support. We gratefully acknowledge the participation of all NIHR Cambridge BioResource volunteers, and thank the Cambridge BioResource staff for their help with volunteer recruitment. We thank members of the Cambridge BioResource SAB and Management Committee for their support of our study and the NIHR Cambridge Biomedical Research Centre for funding. The Cardiovascular Epidemiology Unit is supported by the UK Medical Research Council (G0800270), BHF (SP/09/002), and NIHR Cambridge Biomedical Research Centre. Research in the Ouwehand laboratory is supported by the NIHR, BHF (PG-0310-1002 and RG/09/12/28096) and NHS Blood and Transplant. K.D. is funded as a HSST trainee by NHS Health Education England. M.F. is supported by the BHF Cambridge Centre of Excellence (RE/13/6/30180). A.D., E.L., L.C. and P.F. receive additional support from the European Molecular Biology Laboratory. A.K.S. is supported by an ADA Career Development Award (1-14-CD-17). B.O.B. and R.D.L. acknowledge support from the Deutsche Forschungsgemeinschaft (DFG) and European Federation for the Study of Diabetes, respectively.

## Author contributions

R.D.L., S.B., V.K.R. and D.S.P. designed and supervised the study. M.A.N.D. and D.S.P. performed the experiments. D.S.P, A.E.T., R.L. and S.E. analysed data and performed statistical analyses. All other authors provided samples or analysis tools. D.S.P. wrote the manuscript. All authors read and approved the final manuscript.

## Additional information

**Competing financial interests:** Paul Flicek is a member of the Scientific Advisory Board for Omicia, Inc. All other authors declare no competing financial interests.

