## [Peer Review File · Nature Communications]

Reviewer #1 (Remarks to the Author):

The current study explored the potential for differential DNA methylation (DNAm) in the pathogenesis of T1D. Unfortunately, no significant T1D-associated DMPs in any of the investigated immune cell types, were identified. This suggests that specific cellular pathways are not subject to DNA methylation induced disruption in association with T1D. Although only a single DMP was identified, specific to CD4+ T cells, several T1D-associated DVPs were subsequently identified. These represent 'epigenetic outliers' that often occur in individual twin pairs and cell types. At DVPs, the DNA methylation differences between the T1D twin and its healthy co-twin were found to be comparatively large in many cases. These T1D-associated DVPs are reported as reproducible (in the same cohort); temporally stable (not clear where this finding comes from); not under genetic control; enriched at gene regions responsible for the regulation of gene expression; and located at genes involved in immune cell metabolism and cell cycle.

This is all very nice data, though the major test of the significance of these findings to T1D, namely biological replication in an independent sample, is lacking. Further, though they state the study design removes confounding factors, there is insufficient exploration of other variables, such as medication use or comorbidities, in the analysis.

The pairwise analysis identified cg01674036 as differentially methylated at FDR correction levels. This seems to be downplayed somewhat in the manuscript and it would have been tremendous to see some attempt at further exploring the significance of this, even in a limited replication sample set or in vitro. There is no discussion of the genomic context of this region in terms of ENCODE regulatory elements or

Strengths

- Very well written. Lovely figures
- the use of 52 sets of age matched discordant MZ twin pairs is important, eliminating many of the confounders associated with traditional EWAS. The sample size is sufficient to identify anything meaningful associated with the phenotype
- The use of purified T1D CD4+ T cells, CD19+ B cells, and CD14+CD16- monocytes eliminates many of the cell heterogeneity associated with using whole blood. It is important to acknowledge that each of these cell population is comprised of subsets of cells that may show imbalance between twins, therefore driving any methylation change
- The 450K platform used is the current choice internationally for EWAS analysis. However, it needs to be acknowledged that this represents only a few % of CpG sites in the human genome and is missing many of the recently identified enhancer/regulatory regions identified through ENCODE and other initiatives. Many of these are present on the newer 850K
- The authors are to be applauded for supplementing the 450K data with whole-genome bisulfite sequencing (WGBS-seq) was carried out in four MZ twin pairs,
- >500 million reads per sample resulting in a mean coverage of between 12.6 and 15.1 reads per CpG site. This allowed investigation of over 8.7 million CpGs with a minimum coverage of 10 reads across eight samples. Unfortunately at this depth, there is very little power to detect any meaningful effect size of <10%. Thus the WGBS dataset is of limited value overall. The use of five CpGs or more as a cutoff for identification of DMRs is not fully justified

Weaknesses

- only 406,000 of 485,000 sites were assessed in the current study. This suggests some issues with quality of the available data, potentially masking biologically relevant small effect sizes. Having said that, the fact that MZ twins tend to cluster more closely to their cotwins to unrelated individuals provides some confidence that the data are overall robust
- Clearly, the majority of variation in the dataset is driven by cell type as opposed to some other factor. Thus it would have been beneficial to do MDS and/or PCA analysis on data from each cell type separately, rather than together as shown in Supp Fig 2. Beyond this it is clear that technical

factors account for the majority of the variation in the data

- Several of the DVPs identified appear to be driven in part by a few outlier samples (supp Fig 4), so their direct relationship to T1D per se is questionable. The authors acknowledge these as 'stochastic outlier events that often occur in individual twin pairs and cell types'. It is unclear whether this effect is being driven by a small number of twin pairs within the sample. This is at the heart of the significance, or otherwise, of the study as it necessitates an exploration of a range of potential confounding variables such as medication use, age at diagnosis, severity of disease, age of individual. The authors have attempted to take into account cell heterogeneity, but it is unclear whether this represents heterogeneity of T, B, monocyte cell subtypes (eg. Treg, Th1, Th2, Th17 etc) or some other measure.
- The authors have replicated some of the T1D-associated DNA DVPs using an earlier dataset of CD14+ monocytes and CD4+ T cells from 12 T1D-discordant MZ twin pairs generated 27K platform. These were part of the discovery cohort of this study, but samples were taken and profiled five years earlier. Thus, this is not biological replication, rather technical replication of some of the findings. This cannot even be considered technical replication using a different platform as essentially the 27K and 450K are the same method. Biological replication is lacking.
- The authors screened for DVPs in an independent cohort at birth and they did not replicate, leading to the conclusion that they appear postnatally. This may be the case, but it is equally likely that they are unique to the current population
- There are no data provided on the ages of the different pairs and whether age plays a role in the level of DVPs between pairs. Given that these are thought to arise postnatally, there might be an association between age and the number of DVPs

Reviewer #2 (Remarks to the Author):

An elegant and well written paper on DNA methylations studies in T1D. Numerous strong points: a unique sample of MZ twins discordant for type 1 diabetes, solid statistical analysis with application of new methods, identification of new molecular phenomenon (increased methylation variability), among several others.

Questions/unclear issues/criticisms:

. Application of WGBS (as an addition to the the Illumina 450K array experiment) is evidently underpowered as the sample size was only N=4 twin pairs. Sequencing depth is also rather insufficient for a reliable detection of more subtle DNA methylation differences. What was the point of adding the small scale WGBS when the outcome was predictable? I cannot agree with the statement (line 150) that negative 450K findings were "confirmed in CD4 T cells at genomic loci not covered by the array" or " we also did not find regions of differential DNA methylation in four disease-discordant MZ twin pairs using WGBS-seq (FDR <0.05)".

. In the DVP analysis FDR stringency was very high (0.001). Was there a specific reason for this? Does this mean that the number of DVP loci was very large if a more conventional FDR q (0.05) was applied? Which of the further analyses and findings would be different for different FDR q values?

. Rationale for SNP analysis (lines 173- 174) is not clear. The T1D patients and their co-twins were perfectly matched for genetic background. How can DNA sequence variation account for DVPs in the discordant MZ twin design? What was the point of testing the genotypes?

. Technical replication (lines 187-202) - is it really a technical replication if the samples (27k cohort) were collected 5 years earlier than the 450K cohort? DNA methylation at some loci may have changed unrecognizably over 5 years for biological reasons, and would look like technical "non-replication".

. Analysis of T1D in an unrelated case-control design (lines 204-216) is very valuable but it is not clear why only twin DVPs were analyzed. What kind of differences are detected between unrelated T1D patients and matched controls? Any evidence for DMPs (such were not detected in the twin analysis)? Any new DVPs which were not detected in the twin analysis? I believe the strongest approach would have been testing the two cohorts (MZ twins, unrelated cases-controls) separately and then finding out if the detected loci overlap more than expected by chance only.

. Findings of DVP methylation trends in the genetic elements (promoters, gene bodies; lines 243 - 246) - aren't they similar to the epigenetic changes in tumors? If yes, it would be interesting to know what authors think about this connection. Traditionally, cancer and T1D have been perceived as two very different categories of diseases.

Reviewer #3 (Remarks to the Author):

Summary:

This manuscript reports the results of an epigenomewide association study (EWAS) comparing DNA methylation in twins discordant for type 1 diabetes (T1D). Assays were conducted on flow-sorted CD4+ T cells, CD19+ B cells and monocytes, using the Illumina 450K platform. The authors find few differentially methylated positions (DMPs) but many differentially variable positions (DVPs). The negative DMP finding was confirmed using a limited amount of whole genome bisulfite sequencing (WGBS) data. The positive DVP findings, as well as their cell-type-specificity, were confirmed using different specimens collected earlier from the same subjects, arrayed using the earlier Illumina 27K platform (which uses a biochemistry that is slightly distinct from 450K). They also confirmed the DVPs using data from BLUEPRINT. Additionally, they demonstrate that genetic associations unlikely account for DVP findings. Finally, they investigate functional implications of their findings, highlighting several immune and metabolic pathways.

Review:

This is a high quality study that was conducted and analyzed in an exemplary manner. The findings are very credible, and I have no reservations about them. However, I have a few minor comments about some of the language used in the manuscript and/or interpretations:

1. Line 101: the authors claim that their design eliminates "all major confounding factors in EWASs". Strictly speaking, this is not true, because there are several distinct types of CD4+T cells that the authors have not differentiated. Notably, it is difficult to distinguish Tregs from other types of CD4+ T cells, and Treg homeostatis may be an important mediator of autoimmune disease. I might soften the language just a little bit.

2. In line with the previous comment, I find it interesting that the authors' central functional finding concerns FOXP1, which seems integral to differentiation of T-cells (e.g. <http://www.ncbi.nlm.nih.gov/pmc/articles/PMC2810984/>). To me, this suggests that their data are consistent with heterogeneity in CD4+ T cell subpopulations in a manner that disregulates self/non-self discrimination, which thought to be key in T1D pathogenesis. I would ask the authors to say more about this in the Discussion.

3. Line 180: "We found that T1D-associated DVPs (FDR < 0.001) were depleted at meQTLs..." If I understand correctly, meQTLs did not overlap with DVPs. To me this also seems consistent with my assertion in the previous comment, because meQTLs would dictate genetic predisposition to specific immune patterns within normal variation, while the DVPs would then represent downstream dysregulation in subtle differentiation/activation patterns. More concretely: if the

authors were able to isolate (e.g.) Tregs, I would predict that there would be fewer DVPs and (perhaps) more meQTLs. If possible, it would be interesting if the authors could speculate along this line of thinking in the Discussion.

3. Line 200: "Taken together, we showed that T1D-associated DVPs are reproducible and robust across experimental assays." (Also, the preceding paragraph). This statement is too strong. Given that the authors have not studied independent T1D cases, I would remove the word "reproducible". I still think the findings are very credible, but they have not, in this study, truly been replicated.

4. The legend of Figure 2e refers to B-cells, but I don't see any B-cell results.

We thank all reviewers for their valuable and detailed comments. Please find our responses in blue below. Major changes in the manuscript in response to the reviewers' comments are highlighted in yellow.

Reviewer #1:

The current study explored the potential for differential DNA methylation (DNAm) in the pathogenesis of T1D. Unfortunately, no significant T1D-associated DMPs in any of the investigated immune cell types, were identified. This suggests that specific cellular pathways are not subject to DNA methylation induced disruption in association with T1D. Although only a single DMP was identified, specific to CD4+ T cells, several T1D-associated DVPs were subsequently identified. These represent 'epigenetic outliers' that often occur in individual twin pairs and cell types. At DVPs, the DNA methylation differences between the T1D twin and its healthy co-twin were found to be comparatively large in many cases. These T1D-associated DVPs are reported as reproducible (in the same cohort); temporally stable (not clear where this finding comes from); not under genetic control; enriched at gene regions responsible for the regulation of gene expression; and located at genes involved in immune cell metabolism and cell cycle.

1. This is all very nice data, though the major test of the significance of these findings to T1D, namely biological replication in an independent sample, is lacking. Further, though they state the study design removes confounding factors, there is insufficient exploration of other variables, such as medication use or comorbidities, in the analysis.

We agree with the reviewer that cell type-dependent replication in an independent cohort of identical twins would be ideal. As it took many years to recruit the discovery cohort alone, recruiting a matching replication cohort (with regards to both sample size and cell purification procedure) was not possible within the timelines of this study: We had recruited this sample set in an international effort across twin registries based in the UK, USA, and Germany. To obtain sufficient power for the discovery stage, we were unable to set aside any samples for independent replication, and thus focused instead on the assessment of temporal stability of T1D-associated DVPs, their comparison to individuals with limited genetic T1D risk, as well as their functional annotation and interpretation. Similar to early GWASs, we anticipate that future studies will formally replicate our findings in external cohorts. We have rephrased all relevant sections of the manuscript to acknowledge this limitation. Nonetheless, we believe that our findings, discovered through a unique study design and novel analytical approach, provide a substantial advance in the exploration of non-genetic factors in T1D. Furthermore, we discuss the important implications of these findings for the future application of the EWAS approach to elucidate human disease mechanisms.

We have also rephrased the statement that our study eliminates major confounding factors in EWASs, criticized by the reviewer, to: “Importantly, our experimental design reduces the impact

of all major confounding factors in EWASs, due to the profiling of purified, primary cells derived from MZ twins, who share virtually all somatic variation and early-life environmental exposure.”

Finally, we have explored additional variables that could potentially confound the discovery of DVPs. This analysis is described in detail in response to the comments of reviewer 1 #9.

2. The pairwise analysis identified cg01674036 as differentially methylated at FDR correction levels. This seems to be downplayed somewhat in the manuscript and it would have been tremendous to see some attempt at further exploring the significance of this, even in a limited replication sample set or in vitro.

We thank the reviewer for encouraging us to explore the relevance of the identified DMP further. We have now prepared a regional plot, illustrating the DNA methylation levels at the CpGs surrounding cg01674036 across all individuals stratified for T1D status, and have annotated the genomic locus harboring cg01674036 using cell type-specific chromatin state maps provided by the Roadmap Epigenomics project. We found that the CpG site overlaps with an active gene regulatory region in T cells. Chromatin interaction data further indicated that this regulatory region physically interacts with the promoter region of the gene *DDIT4* (also known as *REDD1*). Our extended literature search revealed that *DDIT4* functions as an inhibitor of the mammalian target of rapamycin complex 1 (mTORC1). Activation of mTORC1 is controlled by anabolic hormones including insulin. Strikingly, the mTOR pathway has also been implicated in the gene network analysis of T1D-associated DVPs (Fig. 4f). These new data add confidence that the discovered DMP indeed exerts a functional role in T1D. We have added these findings to the revised version of the manuscript (Fig. 2).

Fig. 2 | Assessment of the functional significance of the T1D-associated DMP cg01674036. (a) QQ plot for the identification of differentially methylated positions (DMPs) between T1D-discordant MZ twin pairs in different immune effector cell types. Only the DMP cg01674036 reached genome-wide significance in T cells, with $P = 2.2 \times 10^{-9}$ (false-discovery rate (FDR)-corrected $P = 9.1 \times 10^{-4}$) and a mean DNA methylation difference of 2.3%. (b) Regional plot of the locus harboring the T cell-specific DMP cg01674036. The statistically significant DMP is indicated with a black arrow. Data points represent the DNA methylation β -values (y-axis) at the indicated CpGs (x-axis) in one individual. For each CpG site, we calculated the mean DNA methylation value (indicated with a larger data point). Every CpG site is annotated with regards to epigenomic feature and gene element using the 450K array annotation manifest. (c) Annotation of the genomic locus using epigenomic reference datasets. The genomic locus on chromosome 10q22.1

(position = 74,028,000–74,100,000; genome build = hg19) harboring the DMP cg01674036 (chr10:74,058,002) is shown using the WashU Epigenome Browser v40.0.0 (<http://epigenomegateway.wustl.edu/browser/>). The T1D-associated DMP is located at a CpG island (indicated with a red arrow). A total of 16 epigenomic reference tracks provided by the Roadmap Epigenomics project are displayed. Specifically, we show both the primary and imputed chromatin state maps in eight distinct primary T cell populations. The highlighted CpG island overlaps with an active transcription start site (red) or enhancer (orange/yellow) in all available T cell populations. In addition, H3K4me3 ChIA-PET data in the lymphoblastoid cell line GM12878 revealed a long-range chromatin interaction between the active regulatory element and the gene promoter region of *DDIT4*. Abbreviations: CGI=CpG island; IGR=intergenic region; ChIA-PET=chromatin interaction analysis by paired-end tag sequencing.

3. There is no discussion of the genomic context of this region in terms of ENCODE regulatory elements.

We have now performed enrichment analyses with regards to cell type-specific gene regulatory features, specifically chromatin state maps. We used the core 15-state ChromHMM model based on five chromatin marks across 127 epigenomes, including 111 generated by the Roadmap Epigenomics project and 16 generated by the ENCODE project. For all three immune cell types, we found cell type-specific enrichment of T1D-associated hypermethylated DVPs at the chromatin state ‘Active TSS’ (‘TssA’) as well as depletion at ‘Enhancers’ (‘Enh’). These results corroborate our findings based on the Illumina 450K array annotation manifest (Fig. 4). We have added these additional results to the revised manuscript (Supplementary Fig. 7).

Supplementary Fig. 7 | Enrichment of T1D-associated DVPs at reference chromatin states.

(a) Enrichment of DVPs at which the DNA methylation level is increased (hypermethylated; $\Delta\beta > 0$) in T1D twins compared to their healthy co-twins, at 15 distinct chromatin states. We selected epigenomes generated in primary cells derived from peripheral blood that correspond to the cell types investigated here. The enrichment is calculated in relation to all 450K array probes that passed quality control. (b) The same analyses as shown in panel (a), but for DVPs at which the DNA methylation level was reduced (hypomethylated; $\Delta\beta < 0$) in T1D twins. Abbreviations: TssA=active TSS; TssAFlnk=flanking active TSS; TxFlnk=transcription at 5' and 3'-end; Tx=strong transcription; TxWk=weak transcription; EnhG=genic enhancers; Enh=enhancers; ZNF/Rpts=ZNF genes and repeats; Het=heterochromatin; TssBiv=bivalent/poised TSS; BivFlnk=flanking bivalent TSS/enhancer; EnhBiv=bivalent enhancer; ReprPC=repressed Polycomb; ReprPCWk=weak repressed Polycomb; Quies=quiescent/low.

Strengths:

- Very well written. Lovely figures.

- The use of 52 sets of age matched discordant MZ twin pairs is important, eliminating many of the confounders associated with traditional EWAS. The sample size is sufficient to identify anything meaningful associated with the phenotype.
- The use of purified T1D CD4+ T cells, CD19+ B cells, and CD14+CD16- monocytes eliminates many of the cell heterogeneity associated with using whole blood. It is important to acknowledge that each of these cell population is comprised of subsets of cells that may show imbalance between twins, therefore driving any methylation change.

We thank the reviewer for the positive feedback on our study design and manuscript.

5. The 450K platform used is the current choice internationally for EWAS analysis. However, it needs to be acknowledged that this represents only a few % of CpG sites in the human genome and is missing many of the recently identified enhancer/regulatory regions identified through ENCODE and other initiatives. Many of these are present on the newer 850K.

We fully agree with the reviewer, and had already noted the limitations of the Illumina 450K array platform in the submitted version of the manuscript: “The 450K array platform has a fixed set of CpG sites, covering less than 2% of all annotated CpGs. While this platform is scalable to large sample sizes, the complementary application of sequencing-based approaches is required to comprehensively capture disease-associated DNA methylation loci on a genome-wide level.”

We have extended the Discussion section of the revised manuscript to also note the availability of the new 850K array platform, which may enable future discoveries of additional DMPs, DVPs, and DMRs in the context of T1D and other conditions: “It is possible that T1D-associated DMPs and DMRs could be discovered in much larger cohorts or more highly selected cell populations using either Infinium arrays or bisulfite sequencing. In particular, the recent availability of the Illumina Infinium MethylationEPIC BeadChip covering over 850,000 CpG sites, of which many are located at enhancer regions identified by the ENCODE and FANTOM5 projects, may allow for discovery of additional T1D-associated loci.”

6. The authors are to be applauded for supplementing the 450K data with whole-genome bisulfite sequencing (WGBS-seq) was carried out in four MZ twin pairs. >500 million reads per sample resulting in a mean coverage of between 12.6 and 15.1 reads per CpG site. This allowed investigation of over 8.7 million CpGs with a minimum coverage of 10 reads across eight samples. Unfortunately, at this depth, there is very little power to detect any meaningful effect size of <10%. Thus the WGBS dataset is of limited value overall. The use of five CpGs or more as a cutoff for identification of DMRs is not fully justified.

To address the reviewer’s concern, we have provided our power calculation for the WGBS-seq experiments: To determine the power to detect DMRs between T1D and control twins, we simulated the potential read distribution using a binomial distribution. We set the probability of the control samples to 0.5 (i.e. 50% methylated) and the probability of the case samples to $0.5 + d$, where d is the difference in DNA methylation between case and control; for example, $d = 0.3$ represents a 30% DNA methylation difference. As methylation dynamics can be stable at a given genomic locus, we defined the region at which a DNA methylation difference is consistently

observed to contain at least five CpGs. Note that the larger the region of differential methylation, the more power there is to detect it. We set the coverage at each CpG to ten reads. For each simulation, we counted the number of methylated and unmethylated reads in four cases and four controls across these five CpGs. We combined the results into a single 2×2 count matrix and calculated the P -value based on a chi-squared test. We repeated this analysis 1,000 times to calculate how many times we find a significant P -value, and hence, to estimate the power of the experiment. This revealed that there is 70.9% and 94.3% power to detect DMRs with a 30% and 35% DNA methylation difference, respectively.

While we agree with the reviewer that the sequencing depth is insufficient to detect DNA methylation differences at single CpG sites, it has been shown that large continuous blocks of hyper- and hypo-methylation can be detected with a coverage as low as 4x (Hansen et al (2011). *Nature Genet.* 43, 768-775; Hansen et al (2012). *Genome Biol.* 13, R83).

The key finding of this analysis was that there are no DMRs of large effect size genome-wide. This is an important statement, as the 450K array only covers 2% of all CpG sites. We have now rephrased the paragraph to reflect the limitation with regards to statistical power: “This analysis was sufficiently powered to detect differentially methylated regions (DMRs) that consist of at least five CpGs and exhibit a mean DNA methylation difference of >30% at an FDR of <0.05. We did not identify such DMRs to be associated with T1D, irrespective of FDR-values.”

Weaknesses:

7. Only 406,000 of 485,000 sites were assessed in the current study. This suggests some issues with quality of the available data, potentially masking biologically relevant small effect sizes. Having said that, the fact that MZ twins tend to cluster more closely to their co-twins to unrelated individuals provides some confidence that the data are overall robust.

For our analysis approach, we applied conservative thresholds to ensure highest data quality for analysis and biological interpretation (see Methods section). As the sample set for this study was large (n=302 samples), this resulted in removal of a considerable number of probes. As the reviewer previously noted, the resulting data shows excellent quality metrics, as shown in Supplementary Fig. 2.

8. Clearly, the majority of variation in the dataset is driven by cell type as opposed to some other factor. Thus it would have been beneficial to do MDS and/or PCA analysis on data from each cell type separately, rather than together as shown in Supp. Fig. 2. Beyond this it is clear that technical factors account for the majority of the variation in the data.

We have now also performed MDS analysis for all cell types separately, as shown below.

We disagree with the reviewer's comment that technical factors account for the majority of variation. As shown in Supplementary Fig. 2c, the largest component of variation is due to cell type identity (PC1 and PC2), cellular heterogeneity (PC2), and twin pair and donor identity (PC3 and PC4). In addition, we have now explored additional variables that could potentially confound the discovery of DVPs. These results are discussed in response to reviewer 1 #9.

9. Several of the DVPs identified appear to be driven in part by a few outlier samples (Supp. Fig. 4), so their direct relationship to T1D per se is questionable. The authors acknowledge these as 'stochastic outlier events that often occur in individual twin pairs and cell types'. It is unclear whether this effect is being driven by a small number of twin pairs within the sample. This is at the heart of the significance, or otherwise, of the study as it necessitates an exploration of a range of potential confounding variables such as medication use, age at diagnosis, severity of disease, age of individual. The authors have attempted to take into account cell heterogeneity, but it is unclear whether this represents heterogeneity of T, B, monocyte cell subtypes (eg. Treg, Th1, Th2, Th17 etc.) or some other measure.

We fully agree with the reviewer that it is critical to explore, and account for, confounding variables. We had already assessed confounding due to cellular heterogeneity. For this analysis, we used cell purity values as quantified by FACS. This approach assessed the proportion of major leukocyte cell types, specifically CD4⁺ T cells, CD19⁺ B lymphocytes, and CD14⁺ monocytes. We have rephrased the text in the Results section and legend of Supplementary Fig. 5 to clarify that major leukocyte cell populations were analyzed.

In addition, we have revised and improved our approach to assess cellular confounding, and extended it to include additional available variables as suggested by the reviewer. These variables now also include age at T1D diagnosis, age at sample collection, medication use (e.g. statins and thyroxine), as well as presence of other autoimmune diseases (i.e. thyroiditis, as characterized by thyroid peroxidase autoantibodies). To this end, we first calculated the fraction of DVPs in T1D twins showing a significant deviation from the healthy co-twins. Then, we correlated this fraction of deviating DVPs with the different variables. We did not identify a statistically significant correlation with any of the available variables for each T1D case and cell type (Supplementary Fig. 5). Finally, we emphasize that we excluded twins who were pregnant and twins with significant co-morbidities including severe macrovascular and microvascular complications of diabetes.

Supplementary Fig. 5 | Assessment of potential confounding factors for DVP discovery. For each cell type, we selected the DVPs that were found to be hypervariable in T1D twins (FDR <0.001). We then estimated the mean and standard deviation (SD) in DNA methylation across the healthy twins. We used these estimates of mean and SD to normalize the β -valued data matrix for all samples using a z-score. To estimate deviations from the healthy co-twins, one-tailed P -values for each T1D-associated DVP were calculated. Statistically significant deviations were defined as $P < 0.001$. Finally, the fraction of DVPs in T1D twins exhibiting a significant deviation from the healthy co-twins was correlated with potential confounding variables: (a) age of T1D twins at disease diagnosis; (b) age of T1D twins at sample collection; (c) cell purity of samples as quantified by FACS; (d) statin treatment in T1D twins; (e) thyroxine treatment in T1D twins; (f) other

medication use; (g) presence of thyroiditis as characterized by thyroid peroxidase autoantibodies. The three columns correspond to the cell types: B cells (left; green data points), T cells (middle; red), and monocytes (right; blue).

10. The authors have replicated some of the T1D-associated DNA DVPs using an earlier dataset of CD14⁺ monocytes and CD4⁺ T cells from 12 T1D-discordant MZ twin pairs generated 27K platform. These were part of the discovery cohort of this study, but samples were taken and profiled five years earlier. Thus, this is not biological replication, rather technical replication of some of the findings. This cannot even be considered technical replication using a different platform as essentially the 27K and 450K are the same method. Biological replication is lacking.

It is important to note that while these twins were part of the discovery cohort of an earlier study that used 27K arrays, for the study presented here, we re-called these twins, drew fresh blood, sorted cells, extracted DNA, and profiled again the cell type-specific DNA methylation patterns. DNA methylation marks are re-established during cell mitosis, and therefore, observing the same epigenetic patterns across two time points could be considered as biological replication, rather than mere technical replication. The Illumina 27K and 450K arrays are indeed based on the same BeadChip technology, but the additional CpG content is measured by a distinct assay (Infinium II instead of Infinium I).

Nonetheless, we absolutely agree with the reviewer that replication of our findings in an independent cohort is critical. We have rephrased the manuscript to emphasize that our analyses aimed to assess temporal stability rather than biological replication, and added the notion that yet more research is needed to confirm our findings. See also reply to comments by reviewer 1 #1.

11. The authors screened for DVPs in an independent cohort at birth and they did not replicate, leading to the conclusion that they appear postnatally. This may be the case, but it is equally likely that they are unique to the current population.

We agree with the reviewer that it may be conceivable that the identified DVPs are unique to the study cohort. However, our twin cohort and the DiPiS cohort, a population-based prospective study of T1D in Swedish children from which the cord blood samples were obtained, have each been extensively characterized (<http://www.ncbi.nlm.nih.gov/pubmed/?term=DiPiS>). To date, there has not been any evidence that either represent a unique cohort in any respect, as all observations on the diabetic twins have been confirmed in cohorts of diabetic singletons including the Swedish diabetic children.

12. There are no data provided on the ages of the different pairs and whether age plays a role in the level of DVPs between pairs. Given that these are thought to arise postnatally, there might be an association between age and the number of DVPs.

We agree with the reviewer that the age of T1D onset across the study cohort consisting of T1D-discordant MZ twins could be a relevant factor confounding DVP discovery. We have tested this hypothesis, and discuss the results in response to #9.

Reviewer #2:

An elegant and well written paper on DNA methylations studies in T1D. Numerous strong points: a unique sample of MZ twins discordant for type 1 diabetes, solid statistical analysis with application of new methods, identification of new molecular phenomenon (increased methylation variability), among several others.

We thank the reviewer for the positive feedback on our manuscript.

Questions/unclear issues/criticisms:

1. Application of WGBS (as an addition to the the Illumina 450K array experiment) is evidently underpowered as the sample size was only N=4 twin pairs. Sequencing depth is also rather insufficient for a reliable detection of more subtle DNA methylation differences. What was the point of adding the small scale WGBS when the outcome was predictable? I cannot agree with the statement (line 150) that negative 450K findings were "confirmed in CD4 T cells at genomic loci not covered by the array" or "we also did not find regions of differential DNA methylation in four disease-discordant MZ twin pairs using WGBS-seq (FDR <0.05)".

We agree with the reviewer that there is limited statistical power to detect DMRs with modest effect sizes (see further details and power calculation in response to reviewer 1 #6). We would like to explain the rationale for this analysis: The 450K array merely covers 2% of all CpG sites. Our WGBS-seq analysis in four disease-discordant MZ twin pairs indicates that there are no DMRs of large effect size genome-wide.

We have now rephrased the paragraph to reflect the limitation with regards to statistical power, and removed the two statements criticized by the reviewer: "This analysis was sufficiently powered to detect differentially methylated regions (DMRs) that consist of at least five CpGs and exhibit a mean DNA methylation difference of >30% at an FDR of <0.05. We did not identify such DMRs to be associated with T1D, irrespective of FDR-values. In conclusion, with the exception of the DMP cg01674036, we did not identify mean DNA methylation differences between T1D twins and their healthy co-twins in any of the three immune cell types using the 450K array platform (Fig. 3a). At genomic loci not covered by the array, results based on WGBS-seq data indicate that mean DNA methylation differences of large effect size are unlikely to exist."

2. In the DVP analysis FDR stringency was very high (0.001). Was there a specific reason for this? Does this mean that the number of DVP loci was very large if a more conventional FDR q (0.05) was applied? Which of the further analyses and findings would be different for different FDR q values?

The reviewer makes an excellent point. One might even argue that we were not stringent enough and that we ought to have used a Bonferroni-corrected threshold. However, all main conclusions were generally unchanged, when we applied this significance threshold. While the FDR threshold for the analysis of differential variability appears to be stringent, we note that it is nevertheless much less stringent than using a Bonferroni-corrected threshold. Thus, even with a cut-off of FDR <0.001, we do expect a number of features to be false positives. We do not advise further relaxing

the FDR threshold, because iEVORA re-ranks significant DVPs according to a t-statistic. To keep the number of false positives as small as possible, avoiding any impact on the top-ranked features, a stringent FDR of <0.001 was used. We have added a justification for the more stringent FDR threshold in the Methods section of the revised manuscript.

3. Rationale for SNP analysis (lines 173-174) is not clear. The T1D patients and their co-twins were perfectly matched for genetic background. How can DNA sequence variation account for DVPs in the discordant MZ twin design? What was the point of testing the genotypes?

We thank the reviewer for making an important point. Indeed, by studying identical twins who are matched for most genetic variants (albeit not all), DNA sequence variants are unlikely to account for most DVPs. However, it should be noted that post-zygotic mutations can occur and give rise to somatic mosaicism (Castillo-Fernandez et al (2014). *Genome Med.* 6, 60). As the statistical approach of identifying DVPs is novel, and the concept of DNA methylation variability in the context of common diseases other than cancer has not been explored, we wanted to provide conclusive evidence that genetic variation is not a main determinant of DVPs.

4. Technical replication (lines 187-202) - is it really a technical replication if the samples (27k cohort) were collected 5 years earlier than the 450K cohort? DNA methylation at some loci may have changed unrecognizably over 5 years for biological reasons, and would look like technical "non-replication".

As mentioned in the response to reviewer 1 #10, it is important to note that while these twins were part of the discovery cohort of an earlier study that used 27K arrays, for the study presented here, we re-called these twins, drew fresh blood, sorted cells, extracted DNA, and profiled again the cell type-specific DNA methylation patterns. We agree with the reviewer that DNA methylation patterns can change over time; therefore, our analyses that use fresh DNA samples cannot be classed as biological replication, but should be considered as assessment of temporal stability of the discovered DVPs. We have rephrased the manuscript accordingly, and added the notion that more research is needed to further validate our findings in external cohorts.

5. Analysis of T1D in an unrelated case-control design (lines 204-216) is very valuable but it is not clear why only twin DVPs were analyzed. What kind of differences are detected between unrelated T1D patients and matched controls? Any evidence for DMPs (such were not detected in the twin analysis)? Any new DVPs which were not detected in the twin analysis? I believe the strongest approach would have been testing the two cohorts (MZ twins, unrelated cases-controls) separately and then finding out if the detected loci overlap more than expected by chance only.

The reviewer has raised a good point. Crucially, a twin design study allows for the exclusion of specific confounding factors, e.g. genetic variation, age, and *in utero* environmental effects. The reviewer is correct that one would expect to observe the DVPs derived from the twin study to also appear in an unmatched case-control cohort. However, we did not have access to an unmatched case-control cohort, we only had access to an independent set of healthy controls. This set of healthy controls is still useful as it allows us to test the hypothesis whether the identified DVPs from the twin study exhibit more variability in the twin cases compared to this external set of healthy

controls. This was indeed the case, as reported in the Results section and shown in Supplementary Fig. 6.

We could have indeed performed two separate analyses of differential variability, i.e. one comparing our twin cases to their healthy twins, and another comparing our twin cases to the unrelated healthy controls, and then assessing the overlap. Nonetheless, we preferred an approach for which we first use the discordant MZ twin cohort to discover DVPs and then use the unrelated healthy controls as a means of further testing the significance of the identified DVPs.

6. Findings of DVP methylation trends in the genetic elements (promoters, gene bodies; lines 243-246) - aren't they similar to the epigenetic changes in tumors? If yes, it would be interesting to know what authors think about this connection. Traditionally, cancer and T1D have been perceived as two very different categories of diseases.

We think that there is a simple interpretation to these consistent trends at genetic elements: If we aim to identify DNA methylation changes between disease cases and controls (i.e. DMPs or DVPs), it is apparent that such alterations would have to exhibit the same directional pattern irrespective of disease. For example, if our EWAS had identified a significant DNA methylation change at a CpG island, and the respective CpGs were found to be unmethylated in healthy controls, then these CpGs would have to be hypermethylated in cases in order to represent a significant DNA methylation change. This is because reduced levels of DNA methylation cannot be detected at a locus that is normally unmethylated.

The specific set of CpG islands that exhibit hypermethylation identified in our EWAS for type 1 diabetes may be very different to the ones that show hypermethylation in cancer. The enrichment analysis presented in Fig. 4 does not investigate specific CpG islands or gene bodies, it only tests for overall enrichment.

Reviewer #3:

This manuscript reports the results of an epigenomewide association study (EWAS) comparing DNA methylation in twins discordant for type 1 diabetes (T1D). Assays were conducted on flow-sorted CD4+ T cells, CD19+ B cells and monocytes, using the Illumina 450K platform. The authors find few differentially methylated positions (DMPs) but many differentially variable positions (DVPs). The negative DMP finding was confirmed using a limited amount of whole genome bisulfite sequencing (WGBS) data. The positive DVP findings, as well as their cell-type-specificity, were confirmed using different specimens collected earlier from the same subjects, arrayed using the earlier Illumina 27K platform (which uses a biochemistry that is slightly distinct from 450K). They also confirmed the DVPs using data from BLUEPRINT. Additionally, they demonstrate that genetic associations unlikely account for DVP findings. Finally, they investigate functional implications of their findings, highlighting several immune and metabolic pathways.

This is a high quality study that was conducted and analyzed in an exemplary manner. The findings are very credible, and I have no reservations about them. However, I have a few minor comments about some of the language used in the manuscript and/or interpretations:

We are grateful for the positive feedback on our manuscript.

1. Line 101: the authors claim that their design eliminates "all major confounding factors in EWASs". Strictly speaking, this is not true, because there are several distinct types of CD4+T cells that the authors have not differentiated. Notably, it is difficult to distinguish Tregs from other types of CD4+ T cells, and Treg homeostatis may be an important mediator of autoimmune disease. I might soften the language just a little bit.

We thank the reviewer for this comment. We fully agree that subpopulations of isolated cell populations may confound discovery of genuine disease-associated DNA methylation changes, and had already noted this in the Discussion section: "[...] it may also be possible that rare subpopulations of the three immune cell types, such as regulatory T cells (CD25⁺FOXP3⁺ cells) or T helper 17 cells (TH₁₇ cells), harbor epigenetic signals that remain undetected in the broad population of CD4⁺ T cells."

While we believe to have made a substantial improvement with regards to study design in comparison to conventional EWAS design (i.e. case-control studies in whole blood from unrelated individuals), we have now rephrased this statement to: "Importantly, our experimental design reduces the impact of all major confounding factors in EWASs, due to the profiling of purified, primary cells derived from MZ twins, who share virtually all somatic variation and early-life environmental exposure."

2. In line with the previous comment, I find it interesting that the authors' central functional finding concerns FOXP1, which seems integral to differentiation of T-cells (e.g. <http://www.ncbi.nlm.nih.gov/pmc/articles/PMC2810984/>). To me, this suggests that their data are consistent with heterogeneity in CD4+ T cell subpopulations in a manner that dysregulates self/non-self-discrimination, which thought to be key in T1D pathogenesis. I would ask the authors to say more about this in the Discussion.

We thank the reviewer for highlighting this interesting point. We have now extended the Results and Discussion sections to highlight the important role of FOXP1 in B cell, T cell, and monocyte differentiation.

Results section: "*FOXP1* encodes forkhead box P1, an important transcriptional regulator of B cell, T cell, and monocyte differentiation. Recent studies in mice also demonstrated that Foxp1 is essential for islet α -cell proliferation and function, and plays a key role in the regulation of systemic glucose homeostasis."

Discussion section: "Specifically, by integrating T1D-associated DVPs with gene regulatory circuits in CD19⁺ B cells, we pinpointed key transcriptional regulators such as NRF1 and FOXP1 (Fig. 4f), and pathways such as mTOR signaling (Supplementary Table 3). Indeed, the same

signaling pathways have been implicated in differentiation, proliferation, and metabolism of both T cells and monocytes (Feng et al (2010). *Blood* 115, 510-518; Araki et al (2009). *Nature* 460, 108-112; Cheng et al (2014). *Science* 345, 1250684; Ray et al (2015). *Immunity* 43, 690-702). For example, deletion of *Foxp1* in naïve CD8⁺ T cells leads to activation of the mTOR signaling cascade (Wei et al (2016). *J Immunol.* 196, 3537-41), indicating a relationship between gene modules 1 and 3 of the regulatory network we identified here (Fig. 4f). Therefore, DVPs could modulate disease activity through the regulation of immune effector cell gene expression either before or after the induction of the disease process.”

3. Line 180: "We found that T1D-associated DVPs (FDR < 0.001) were depleted at meQTLs..." If I understand correctly, meQTLs did not overlap with DVPs. To me this also seems consistent with my assertion in the previous comment, because meQTLs would dictate genetic predisposition to specific immune patterns within normal variation, while the DVPs would then represent downstream dysregulation in subtle differentiation/activation patterns. More concretely: if the authors were able to isolate (e.g.) Tregs, I would predict that there would be fewer DVPs and (perhaps) more meQTLs. If possible, it would be interesting if the authors could speculate along this line of thinking in the Discussion.

Indeed, we found DVPs to be depleted at meQTLs, suggesting DNA sequence variants are unlikely to account for most DVPs. This result was expected, because our study design using identical twins reduced confounding due to genetic variation. Specifically, due to the homogeneous genetic background, the genetic risk of developing T1D is expected to be equivalent.

We agree with the reviewer, and had already noted in the Discussion section, that DVPs could modulate disease activity through the regulation of immune effector cell gene expression either before or after the induction of the disease process. However, it also remains possible that some T1D-associated DVPs may result from disease-associated metabolic disturbances. Further research is needed to prove whether DVPs are causally implicated in T1D pathogenesis. To this end, the study of T cell subpopulations, including regulatory T cells (as noted in the Discussion section), are paramount.

4. Line 200: "Taken together, we showed that T1D-associated DVPs are reproducible and robust across experimental assays." (Also, the preceding paragraph). This statement is too strong. Given that the authors have not studied independent T1D cases, I would remove the word "reproducible". I still think the findings are very credible, but they have not, in this study, truly been replicated.

We fully agree with the reviewer, and have rephrased the corresponding sections in the revised version of the manuscript. Please also refer to the response to reviewer 1 #10 and reviewer 2 #4.

5. The legend of Figure 2e refers to B-cells, but I don't see any B-cell results.

Panels a–c of Fig. 3 (Fig. 2 in the previous version) discuss results obtained for all three cell types. For assessment of temporal stability and evaluation of T1D-associated DVPs compared to individuals with limited genetic T1D risk (panels d, e), we analyzed DNA methylation profiles obtained from T cells and monocytes.

Reviewer #1 (Remarks to the Author):

The authors have taken on board all comments and have done their best to address them using additional analysis of the same dataset. Though this still leaves a major question in regards to the generalisability of the findings, in the absence of getting the data out to the broader community, this cannot be tested. I therefore support publication of the revised manuscript but would like assurance that the raw unprocessed dataset with appropriate subject information will be available for others to attempt replication. In the absence of such assurance, I don't feel the study warrants publication until replicated by the same team in an independent sample

Reviewer #3 (Remarks to the Author):

The authors have addressed all of my prior comments.

Reviewer #1:

The authors have taken on board all comments and have done their best to address them using additional analysis of the same dataset. Though this still leaves a major question in regards to the generalisability of the findings, in the absence of getting the data out to the broader community, this cannot be tested. I therefore support publication of the revised manuscript but would like assurance that the raw unprocessed dataset with appropriate subject information will be available for others to attempt replication. In the absence of such assurance, I don't feel the study warrants publication until replicated by the same team in an independent sample.

In compliance with the data access agreement of the IHEC/BLUEPRINT Consortium, all raw data and appropriate subject information had already been deposited in the European Genome-phenome Archive (EGA). We confirm that the accession codes are now online and publicly available.

We have added the following paragraph to the Methods section:

Data availability. All 450K array and WGBS-seq data sets that support the findings of this study have been deposited in the European Genome-phenome Archive (EGA) with the accession code EGAS00001001598 (<https://www.ebi.ac.uk/ega/studies/EGAS00001001598>). We retrieved 450K array data sets of CD14⁺ and CD4⁺ cells from EGA with the accession code EGAS00001001456 (<https://www.ebi.ac.uk/ega/studies/EGAS00001001456>), and 27K array data sets of CD14⁺ and CD4⁺ cells from Gene Expression Omnibus (GEO) with the accession code GSE56606 (<http://www.ncbi.nlm.nih.gov/geo/query/acc.cgi?acc=GSE56606>).

Reviewer #3:

The authors have addressed all of my prior comments.

We thank the reviewer for the positive feedback on our manuscript.